# Electrical resistance of the current collector controls lithium morphology

Solomon T. Oyakhire [1,5], Wenbo Zhang[2,5], Andrew Shin[2], Rong Xu[2], David T. Boyle [3], Zhiao Yu [3], Yusheng Ye [2], Yufei Yang[2], James A. Raiford[1], William Huang[2], Joel R. Schneider [1], Yi Cui [2,4 ✉] & Stacey F. Bent [1 ✉]

The electrodeposition of low surface area lithium is critical to successful adoption of lithium metal batteries. Here, we discover the dependence of lithium metal morphology on electrical resistance of substrates, enabling us to design an alternative strategy for controlling lithium morphology and improving electrochemical performance. By modifying the current collector with atomic layer deposited conductive (ZnO, SnO$_2$) and resistive (Al$_2$O$_3$) nanofilms, we show that conductive films promote the formation of high surface area lithium deposits, whereas highly resistive films promote the formation of lithium clusters of low surface area. We reveal an electrodeposition mechanism in which radial diffusion of electroactive species is promoted on resistive substrates, resulting in lateral growth of large (150 μm in diameter) planar lithium deposits. Using resistive substrates, similar lithium morphologies are formed in three distinct classes of electrolytes, resulting in up to ten-fold improvement in battery performance. Ultimately, we report anode-free pouch cells using the Al$_2$O$_3$-modified copper that maintain 60 % of their initial discharge capacity after 100 cycles, displaying the benefits of resistive substrates for controlling lithium electrodeposition.

[1] Department of Chemical Engineering, Stanford University, Stanford, CA 94305, USA. [2] Department of Materials Science and Engineering, Stanford University, Stanford, CA 94305, USA. [3] Department of Chemistry, Stanford University, Stanford, CA 94305, USA. [4] Stanford Institute for Materials and Energy Sciences, SLAC National Accelerator Laboratory, 2575 Sand Hill Road, Menlo Park, CA 94025, USA. [5] These authors contributed equally: Solomon T. Oyakhire, Wenbo Zhang. ✉email: yicui@stanford.edu; sbent@stanford.edu

Lithium metal batteries present a complex intersection of opportunities and challenges for energy storage. With a gravimetric capacity of 3860 mAh/g, the lithium metal anode holds immense energy storage potential. Unfortunately, lithium undergoes numerous instabilities during electrodeposition—including continuous decomposition of the electrolyte and formation of dendritic structures which pose safety hazards—that render its practical deployment untenable[1–4]. Efforts towards passivating lithium have been focused on the design of stable interphases between lithium and the electrolyte, many of which have been implemented using molecular techniques like electrolyte engineering[5–8] and thin film methods like atomic and molecular layer deposition[9–13]. These methods result in battery performance improvements that are typically associated with the passivation of the lithium–electrolyte interface. While lithium–electrolyte interface plays a key role in stabilizing the battery, there is no clarity on how these widely adopted lithium passivation strategies impact the electrodeposition of lithium at the lithium–copper interface.

Copper is commonly used as a current collector in lithium metal batteries due to its high conductivity, but it binds weakly to Li[14]. This weak binding increases the nucleation overpotential required for the electrodeposition of lithium, resulting in the formation of lithium deposits with small critical nuclei sizes[15]. Small lithium deposits formed during nucleation on bare copper aggregate into high surface area lithium deposits, which leads to issues with rapid electrolyte consumption[16]. Hence, much research activity has been geared towards reducing the surface area of lithium deposits. Most common approaches like electrolyte engineering do not directly address the intrinsic instabilities at the copper current collector surface, resulting in poor electrodeposition at practical cycling conditions. As a result, other approaches such as copper modification strategies are increasingly critical for improving lithium electrodeposition at the substrate. To understand and address the tendency of bare copper substrates to form lithium deposits with high surface area, low nucleation overpotential (lithiophilic) substrates are being explored for promoting the deposition of low surface area lithium. Gold (Au) seeds[17] and numerous graphene derivatives[18–23] have been used as lithiophilic layers that promote the formation of low surface area lithium deposits by forming lithium alloys prior to lithium nucleation. Our previous work showed that the use of lithiophilic, atomic layer deposition (ALD) grown thin films of $TiO_2$ coated on copper resulted in the formation of low surface area lithium clusters via a scalable route[24]. Yet, while the strategy of modifying the surface of copper with lithiophilic materials yields better performing lithium metal batteries, lithiophilic substrates provide only limited improvements in terms of lithium surface area reduction. As a result, to push the performance of lithium metal batteries towards commercial relevance, it is critical to identify new substrate properties that can substantially reduce the surface area of lithium formed during electrodeposition.

In the present work, we identify a key influence in controlling lithium electrodeposition morphology, that of electrical resistance. We study the properties of several copper modification films and expand upon the most common design property—lithiophilicity—to obtain better performing lithium metal batteries. We use ALD to modify copper with sub-10 nm films of $SnO_2$, ZnO, and $Al_2O_3$ and show how the electrical resistance of these films are correlated with lithium morphology. By using ALD, we ensure conformality of films, and by maintaining film thickness below 10 nm, we preserve the high-energy density of lithium metal. We show that the highly resistive $Al_2O_3$-modified copper supports the formation of low surface area lithium deposits while bare copper and copper modified with $SnO_2$ and ZnO promote the formation of high surface area lithium deposits, as illustrated schematically in Fig. 1. We propose that the high resistance of $Al_2O_3$ reduces the available nucleation sites for lithium metal promoting sparse nucleation of lithium deposits, and that the radial diffusion of lithium ions towards the nucleated deposits promotes lateral growth of lithium, resulting in dense, and low surface area lithium deposits. We show that the solid electrolyte interphase (SEI) formed from electrolyte decomposition atop lithium in the presence of each modified copper substrate is chemically and structurally similar, indicating that in this work the changes in lithium morphology stem from differences in substrate properties and not the SEI. We show by scanning electron microscopy (SEM), cryogenic transmission electron microscopy (cryo-TEM), X-ray diffraction (XRD), X-ray photoelectron spectroscopy (XPS), coulometry, and electrical resistivity measurements that the principal difference across substrates is electrical resistance. By cycling the modified substrates in Li||Cu cells using 1 M lithium bis(trifluoromethanesulfonyl)imide (LiTFSI) in 1:1 v:v 1,3 dioxolane:1,2-dimethoxyethane with 1 wt% lithium nitrate additive (this electrolyte denoted as DDN hereafter), we demonstrate the superior performance of the highly resistive $Al_2O_3$-modified copper, reporting an average coulombic efficiency (CE) of 96% over about 400 cycles at a current density of 1 mA/cm² and a capacity of 1 mAh/cm². We also report anode-free pouch cells that maintain 60% of their initial discharge capacity after 100 cycles when cycled using $Al_2O_3$-modified copper. Finally, we demonstrate the generalizability of the $Al_2O_3$-modified copper by showing similar performance and morphology improvements in two other distinct electrolytes—1M lithium hexafluorophosphate (LiPF$_6$) in 1:1 v:v ethylene carbonate:diethylcarbonate with 10% fluoro ethylene carbonate (EC/DEC/FEC), and a state-of-the-art electrolyte, 1M lithium bis(fluorosulfonyl)imide (LiFSI) in fluorinated 1,4-dimethoxylbutane (FDMB)[8]. The similarity in lithium morphology observed across three distinct classes of electrolytes suggests that electrical resistance may serve as a new parameter for improving lithium nucleation and extending cycling performance in lithium metal batteries.

## Results

To probe the reversibility of lithium plating in the presence of each modified copper substrate, we carry out CE tests at a current density of 1 mA/cm² and a capacity of 1 mAh/cm², normalized to the geometric electrode area of Li, in Li||Cu coin cells using 60 μL of DDN electrolyte. Copper substrates modified with 4 nm of ALD films do not show significant performance differences in comparison with bare copper foils, possibly because film coverage is not complete at that thickness (Supplementary Fig. 1). Using thicker films (~8 nm) for the modified copper substrates, significant improvements in cycling reversibility are observed (Fig. 2a). Here we define a metric of cell capacity fade as the cycle index at which each cell declines beyond 90% CE. This metric is tied to the rate of electrolyte consumption in the DDN electrolyte since CE reduces when $LiNO_3$ is consumed. Whereas the bare copper cells show capacity fade after about 41 cycles, cells modified with $SnO_2$ show capacity fade after 84 cycles, and cells modified with ZnO show capacity fade after 118 cycles (Fig. 2a). The cell with bare copper shows capacity fade in multiple cycle index regions indicating that it experiences cycling instabilities associated with the loss and retrieval of lithium due to SEI formation and possibly dead lithium formation. Significantly, the $Al_2O_3$-modified cells show higher cycling stability, reaching a cycle index of 390 at an average CE of 96% before showing signs of capacity fade. Cycling performance is verified using at least two cell replicates (Supplementary Fig. 2). Owing to the performance of $Al_2O_3$-modified copper at 1 mA/cm², we also tested it at

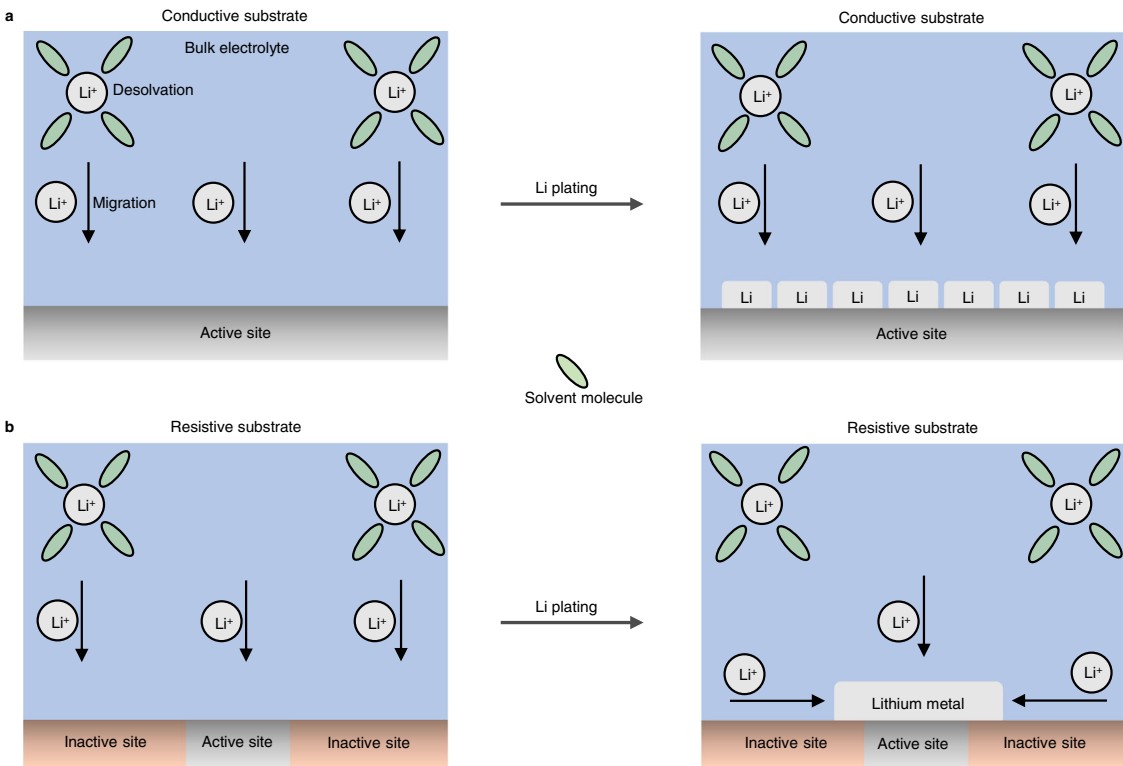

**Fig. 1 Electrical property of ALD-modified copper influences lithium morphology. a** Illustration of lithium electrodeposition on a substrate modified by a conductive ALD film. Here, multiple nucleation sites accompany the formation of lithium deposits with high packing density (high exposed surface area). **b** Illustration of lithium electrodeposition on a substrate modified by a resistive ALD film. Here, few nucleation sites accompany the formation of lithium deposits with low packing density (low exposed surface area). SEI is not shown for simplicity and could vary across substrates.

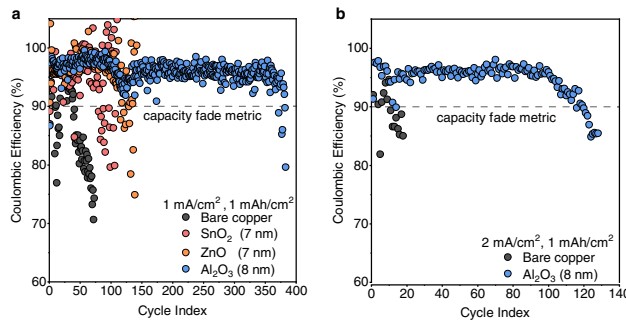

**Fig. 2 Electrochemical performance of ALD-modified cells demonstrated using Li||Cu cells.** All cells were cycled in DDN electrolyte. **a** CE of Li||Cu cells cycled at a current density of $1 \, mA/cm^2$ with an electrodeposition capacity of $1 \, mAh/cm^2$ using bare copper, and copper modified with 7–8 nm of $SnO_2$, $Al_2O_3$, and ZnO. **b** CE of Li||Cu cells cycled at a current density of $2 \, mA/cm^2$ with an electrodeposition capacity of $1 \, mAh/cm^2$ using bare copper, or copper modified with 8 nm of $Al_2O_3$.

$2 \, mA/cm^2$. The $Al_2O_3$-modified cells also show significant improvements over bare Cu at a current density of $2 \, mA/cm^2$, extending cycle life, as determined by our capacity fade metric, from 10 cycles to 120 cycles (Fig. 2b). The large differences in cycling reversibility between the modified versus bare copper substrates may be the result of changes in lithium morphology conferred by the ALD-modified copper substrates, as investigated below.

To understand the effect of the modified substrates on the morphology of lithium, we carry out SEM analysis on Li||Cu cells after the 1st and 50th cycle of plating using a current density of $1 \, mA/cm^2$ and a capacity of $1 \, mAh/cm^2$. The analysis is carried out on bare copper, and on copper current collectors modified with 7 nm of $SnO_2$, 7 nm of ZnO, and 8 nm of $Al_2O_3$.

After the first cycle of plating, the lithium morphology on bare copper presents as characteristic small nuclei of diameter ~2.84 μm (±0.44 μm) with high areal density (Fig. 3a). Here, and in subsequent discussions in this section, any reported nuclei diameter is an average of at least 10 distinct lithium particles, with one standard deviation of that average reported in parentheses. These small nuclei are typically associated with rapid electrolyte consumption, owing to their large, exposed surface area[16,25]. In comparison, the lithium morphology on $SnO_2$ has a long, snake-like structure that appears isolated from the current collector, with approximate diameter of 3.77 μm (±1.36 μm) (Fig. 3b). The tendency for lithium to coalesce into long deposits of this morphology could be an effect of the interfacial energy between $SnO_2$ and lithium, and the apparent electrical isolation of the deposits may explain the quick decline in performance of the $SnO_2$-modified cells. Additional SEM images that show the isolation of Li particles atop $SnO_2$ are presented in Supplementary Fig. 3. Interestingly, ZnO promotes two distinct sizes of nuclei, large and small, with an average diameter of 3.56 μm (±2.3 μm), with the large standard deviation in particle size possibly indicating incomplete coverage of the bare copper by ZnO, since the smaller particle sizes are similar to those formed on the control copper substrates (Fig. 3c). The small nuclei formed on the ZnO-modified Cu would also be expected to lead to quick electrolyte consumption and may be the limiting factor in its electrochemical reversibility. In contrast to all the other substrates, $Al_2O_3$ supports sparse lithium nucleation, with a distinctive Li morphology of aggregated clusters of ~91.47 μm (±43.9 μm) in diameter (Fig. 3d). These aggregated lithium deposits on the $Al_2O_3$-

Top-view morphology of lithium atop all substrates in DDN electrolyte on the 1st cycle of electrodeposition

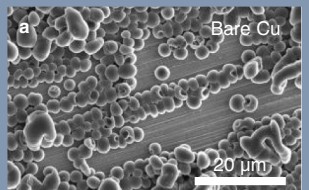 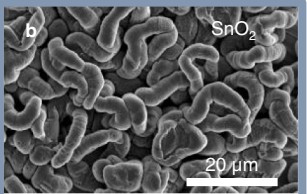 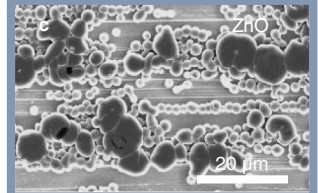 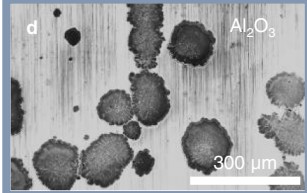

Cross-sectional morphology of lithium atop all substrates in DDN electrolyte on the 1st cycle of electrodeposition

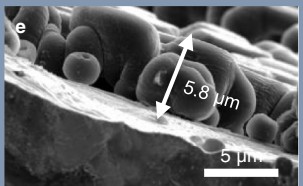 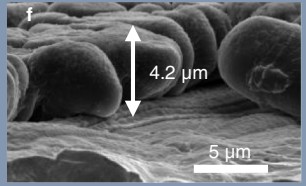 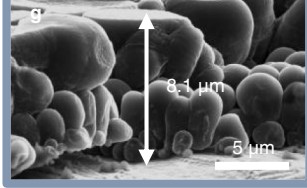 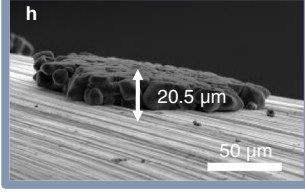

Top-view morphology of lithium atop all substrates in DDN electrolyte on the 51st cycle of electrodeposition

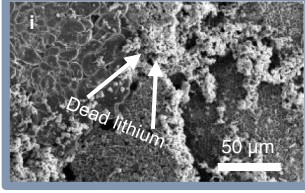 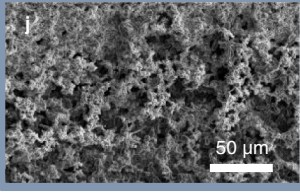 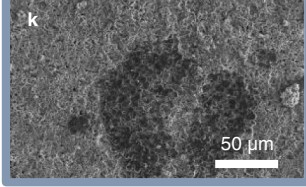 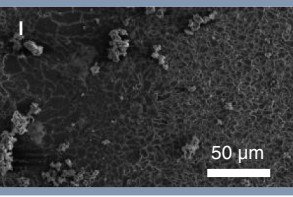

**Fig. 3 Effects of ALD modification on lithium morphology. a–d** Top-view SEM images of lithium deposits formed on bare copper and copper modified with $SnO_2$, ZnO, and $Al_2O_3$, respectively after the first cycle of lithium deposition at 1 mA/cm$^2$ with an electrodeposition capacity of 1 mAh/cm$^2$. **e–h** Cross-sectional SEM images of lithium deposits formed on bare copper and copper modified with $SnO_2$, ZnO, and $Al_2O_3$, respectively, after the first cycle of lithium deposition at 1 mA/cm$^2$ with an electrodeposition capacity of 1 mAh/cm$^2$. **i–l** Top-view SEM images of lithium deposits formed on bare copper and copper modified with $SnO_2$, ZnO, and $Al_2O_3$, respectively, in the 51st cycle of lithium deposition after 50 cycles of electrochemical cycling at 1 mA/cm$^2$ with an electrodeposition capacity of 1 mAh/cm$^2$ in each cycle.

modified copper substrate are expected to consume less electrolyte during cycling, an effect which would explain the large improvement in cycling reversibility of $Al_2O_3$-modified copper shown in Fig. 2a. Low magnification SEM images show that the Li particles captured in Fig. 3a–d are representative (Supplementary Fig. 4), and size distribution analysis obtained using at least 10 distinct Li particles is displayed in Supplementary Fig. 5.

Cross-sectional SEM images after the first cycle of plating further reveal the distinct microstructure of lithium in the presence of ALD-modified substrates. The morphology of plated lithium reveals a vertical thickness of about 5.8 μm in the presence of bare copper (Fig. 3e), 4.2 μm on $SnO_2$ (Fig. 3f), 5–8 μm, depending on the type of nuclei observed, in the case of ZnO (Fig. 3g), and about 20.5 μm in the presence of $Al_2O_3$ (Fig. 3h). A simple calculation confirms that the differences in lithium thickness correspond with the lithium radii and areal densities observed in Fig. 3a–d, since the same quantity of lithium is deposited atop each substrate. The $Al_2O_3$-modified substrate has the lowest areal density of lithium deposits and as such, it has the thickest lithium deposits. It is noteworthy that the microstructure of lithium on $Al_2O_3$ appears more compact and interlocked than does lithium on the other substrates, possibly indicating better contact with the copper foil. Lithium deposits that are in good contact with copper are reportedly more electrochemically retrievable[26,27], consistent with the improved performance observed in the presence of $Al_2O_3$. While there is clear evidence that the different Li morphologies formed on these substrates contribute to differences in performance, it is possible that factors such as galvanic corrosion also influence performance differences on a long-term scale.

After 50 electrochemical cycles, the morphology of lithium on bare copper and all ALD-modified substrates looks similar, with each exhibiting a coalesced structure (Fig. 3i–l). While the

deposits appear coalesced, there is evidence of an accumulation of structures with lighter contrast and more porous morphologies than the freshly plated lithium. The lighter structures can be identified as accumulated SEI or dead, electrochemically irretrievable lithium as has been previously demonstrated[28]. There is a large accumulation of dead lithium on the bare copper, $SnO_2$, and ZnO substrates (Fig. 3i–k), and a relatively small accumulation of those structures on the $Al_2O_3$ cells (Fig. 3l). The relative scarcity of accumulated dead lithium in the $Al_2O_3$-modified cells suggests that the majority of lithium is in electrical contact with the current collector, explaining why lithium electrodeposition and stripping is highly reversible atop $Al_2O_3$. This morphological stability elicited by $Al_2O_3$ is also supported by electrochemical impedance spectroscopy results collected after 50 electrochemical cycles, with the $Al_2O_3$-modified cell showing the least solution resistance and charge transfer resistance across the SEI among the four different substrates (Supplementary Fig. 6 and Supplementary Table 1). Solution resistance represents resistance to ion transport within the liquid electrolyte[29], so the low solution resistance observed with $Al_2O_3$ substrates indicates a reduction in electrolyte consumption, while the low SEI charge transfer resistance is evidence of the lower prevalence of dead lithium. While an inverse relationship between charge transfer resistance and lithium exists, we assume that the lithium particles across all substrates have similar surface areas after the 50th cycle due to their similar SEM morphologies, making our estimates of charge transfer resistance reasonable for identifying the prevalence of dead Li.

To determine the geometrical position of the ALD films used in this study, we carry out XPS on samples with 1 mAh/cm$^2$ of freshly plated lithium, deposited at 1 mA/cm$^2$. If the films used in our study function as nucleation layers, we expect electrodeposition of lithium to occur atop them, but if they function as

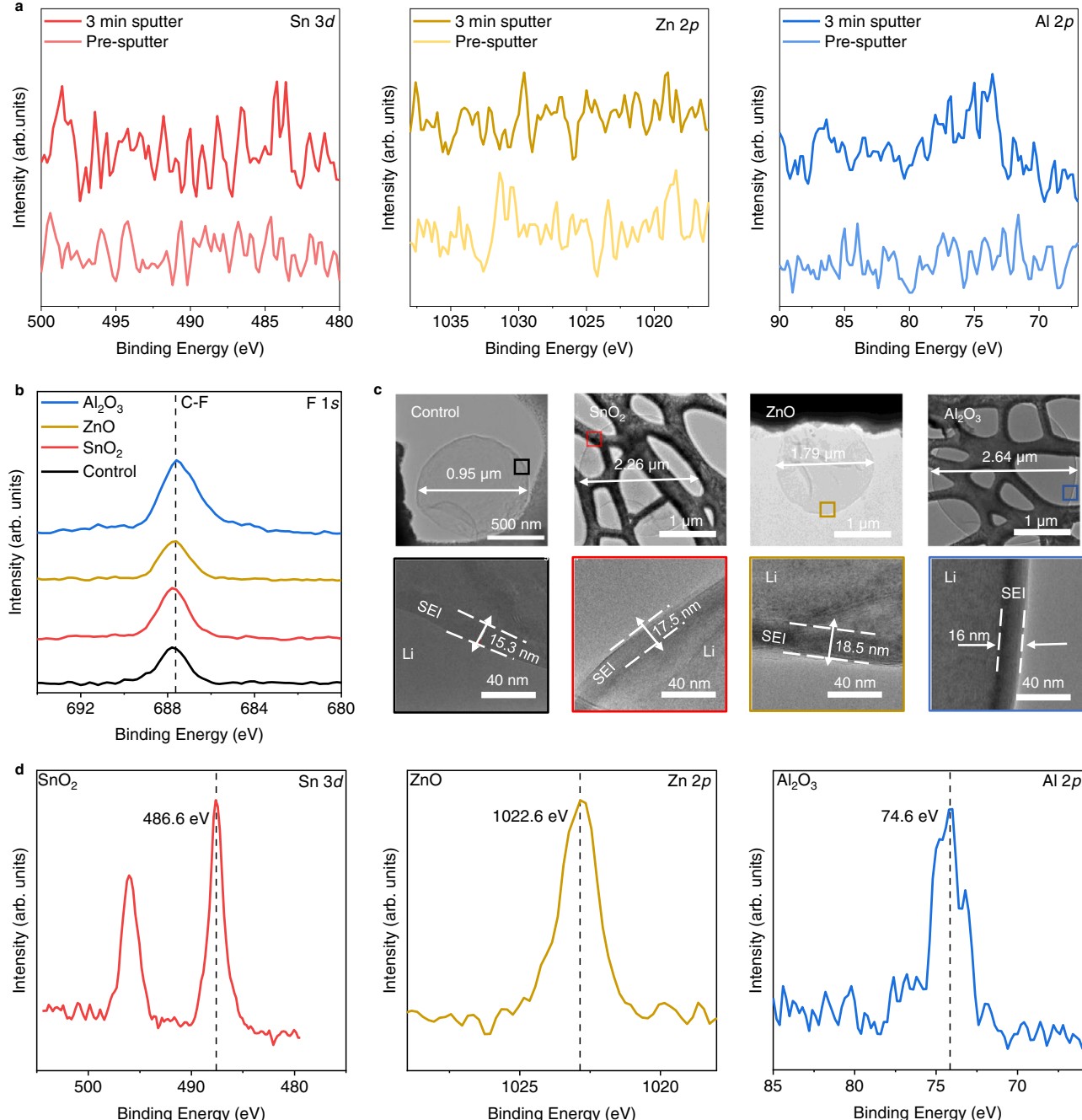

**Fig. 4 Chemical characterization reveals similarities between bare copper and ALD-modified copper. a** High-resolution XPS scan of 1 mAh/cm² of plated lithium deposited onto ALD-modified substrates, before and after 3 min of sputtering carried out at the rate of 4 nm/min calibrated for SiO₂. **b** XPS high resolution scan showing F 1s signals in the SEI formed atop 1 mAh/cm² of lithium deposited on bare copper and copper modified with ALD films. **c** Cryo-TEM images of 0.1 mAh/cm² of lithium plated on bare copper and on ALD-modified copper in DDN electrolyte, showing lithium particle size and SEI thickness. Top row shows the size of lithium particles and the bottom row shows the SEI obtained by magnifying select portions of the images in the top row. The web-like features in the SnO₂ and Al₂O₃ images are segments of the TEM grids. **d** High resolution XPS scan of ALD-modified copper substrates showing characteristic peaks of corresponding ALD film metal centers after potentiostatic holds at 15 mV vs. Li⁺/Li for 8 h.

artificial SEIs lithium is expected to deposit beneath them. Detection by XPS of a nucleation film that sits below a Li over-layer is not expected because the depth sensitivity of XPS is about 5 nm[30], whereas the thickness of 1 mAh/cm² of deposited lithium is ~5 μm. The absence of Al, Sn, and Zn signals prior to and after three minutes of sputtering indicates that lithium is deposited atop the Al₂O₃, SnO, and ZnO nucleation films, respectively (Fig. 4a). By sputtering 0.1 mAh/cm² of Li freshly deposited at 1 mA/cm² on Al₂O₃, we observe a reduction in SEI-specific

elements (C, N, and F) and a gradual increase in Al and Cu signals (Supplementary Fig. 7), indicating that Li deposits atop our ALD films. This observation is in agreement with our previous report in which lithium plated atop TiO₂[24]. We observe a small Al signal after sputtering the surface of Li in Fig. 4a due to the sparsity of lithium deposits formed atop Al₂O₃, for which it is very likely that the area of XPS analysis was near exposed surfaces of Al₂O₃.

The performance of lithium metal batteries and the morphology of lithium deposits are typically correlated with the chemical species that form in the SEI, a solid layer that forms atop lithium because of electrolyte decomposition[31,32]. We examine by XPS the SEI formed on 1 mAh/cm$^2$ of lithium metal freshly deposited on bare copper and on ALD-modified copper at 1 mA/cm$^2$. The high-resolution F 1$s$ peak after 1 min of sputtering (~2 nm in depth calibrated for SiO$_2$) indicates the presence of the same C–F bond across all examined samples (Fig. 4b). And because F is present in only the salt, the C–F bonded species is likely a product of salt decomposition in the electrolyte[33]. Similar observations are found in the O 1$s$ and S 2$p$ high-resolution spectra, with the SEI formed on Li revealing similar bonds between bare-copper and ALD-modified copper samples (Supplementary Fig. 7). All substrates reveal O 1$s$ spectra containing C–O and Li$_2$O, and weak S 2$p$ spectra (Supplementary Fig. 8a, b). This consistency in chemical composition of the SEI despite changes in substrate structure is expected because we used the same electrolyte across substrates and the cells were operated at similar temperatures (~25 °C)[16]. It is noteworthy that we focused on identifying the chemical species in the SEI rather than quantifying them due to the variability in SEI composition across the surface of Li. Instead, we use the presence of similar SEI chemical composition across our substrates to show that they have similar Li–electrolyte interfaces.

While the SEI chemical composition is critical for cell lifetime and performance, its structure and thickness also play a key role in the ease of ion mobility across electrode–electrolyte interfaces. Using fully developed cryogenic-TEM (cryo-TEM) methods[34,35], we preserve the native SEI and examine its structure and thickness on 0.1 mAh/cm$^2$ of lithium, freshly plated at 1 mA/cm$^2$. A relatively small capacity of lithium is used to ensure electron transparency during cryo-TEM analysis. SEI information collected using cryo-TEM complements the chemical information collected from XPS analysis even though Li deposition capacities are different[8]. From Fig. 4c, the cryo-TEM images on the top row reveal the size of the Li deposits on the corresponding substrates, and the images on the bottom row show the SEI thickness at the Li borders, obtained by magnifying outlined portions of the images from the top row. The lithium deposits are distinguished from the corresponding SEIs by differences in image contrast under the TEM beam, as has been previously demonstrated, with the faint circular features in the top view images attributed to Li and the thinner, darker features at the Li-electrolyte interface attributed to SEI[16,35]. The TEM data indicate that the particle size of lithium is largest on the Al$_2$O$_3$-modified substrate (2.64 μm) and smallest on the bare copper substrate (0.95 μm). The trend in particle size of lithium for the different substrates observed by TEM (Fig. 4c) agrees with our SEM data (Fig. 3a–d). In addition, the TEM results indicate that across all substrates, the SEI formed on lithium has thickness between 15 and 18.5 nm and is amorphous, as it does not contain ordered domains (Fig. 4c). These SEI thicknesses are further corroborated by measurements over 10 distinct SEI domains of Li deposited on each substrate, with the average SEI thickness on bare copper, SnO$_2$, ZnO, and Al$_2$O$_3$ being 17.1 nm (±5.6 nm), 16.3 nm (±1.9 nm), 17.7 nm (±5.0 nm), and 17.4 nm (±2.8 nm), respectively (Supplementary Fig. 9). Here, the numbers in parentheses represent one standard deviation of the average SEI thicknesses. The similarity in SEI thickness, composition, and structure for all four substrates indicates that the different lithium morphologies observed on each substrate are not caused by the SEI (the lithium–electrolyte interface). This finding indicates that changes in lithium metal morphology and battery performance observed in this study are not associated with SEI modifications, in contrast to several previous reports[8,36]. Rather, they must be borne from differences

at the interface between Li and the current collector. As such, emphasis should be placed on investigating the differences at the Li–current collector interface.

To understand the lithium–current collector interface, we examine the chemical nature of the substrates just before the onset of lithium nucleation. In our previous study, we reported that TiO$_2$ reacts with lithium to form a Li$_x$TiO$_2$ alloy prior to the onset of electrodeposition[24]. Changes to the current collector prior to nucleation could elicit differences in lithium morphology especially if a lithium alloy is formed on the current collector. To investigate the interface between lithium and the ALD-modified current collectors, we hold the Li||Cu cells at 15 mV, just above the nucleation potential for lithium, for 8 h. Following this voltage hold, we carry out XPS on the ALD-modified copper substrates. From Fig. 4d, it is evident that there is no reaction between Al$_2$O$_3$ and lithium ions prior to electrodeposition because the binding energy of Al 2$p$ (74.6 eV) remains consistent with Al in the bonding environment of Al$_2$O$_3$[37]. This result is not surprising because the lithium and Al$_2$O$_3$ reaction is known to have a high energy barrier at room temperature[38]. Figure 4d also reveals that SnO$_2$ and ZnO do not react with Li prior to nucleation, as indicated by the binding energy of Sn 3$d$ 5/2 and Zn 2$p$ 3/2, which show up at 486.6 and 1022.6 eV, respectively, suggesting the presence of SnO$_2$[39] and ZnO[40]. This finding is surprising because the reaction between lithium and SnO$_2$ or ZnO should be energetically and electrochemically favorable[41]. Using cyclic voltammetry (CV), we find that conversion and alloy reactions occur on the SnO$_2$ and ZnO substrates (Supplementary Fig. 10). However, because the XPS binding energy of Sn and Zn indicate the presence of their corresponding oxides, it is likely that a large fraction of the ALD films do not react with Li ions during the potentiostatic hold at 15 mV. Because there is no strong evidence for direct reaction between Li ions and any of the three metal oxide films, the observed differences in lithium morphology and performance of cells appear to be the result of differences in the intrinsic properties of the ALD films used to modify the copper current collector.

The intrinsic characteristics of a substrate that could govern nucleation include structural, electrical, and chemical properties, and we investigate them in that order. Past reports have shown that the exposed crystallographic facets of substrates impact the adsorption energy of lithium during electrodeposition[14,25]. Using XRD, we probe the copper substrate before and after ALD modification to examine the diffraction peaks of copper. For all four substrates, the diffraction peaks observed are typical of those present in bare copper, with the dominant peaks being (111), (200), and (220) (Supplementary Fig. 11a). This observation indicates that the ALD processing conditions do not significantly alter the structural properties of the underlying copper substrate. This conclusion is also buttressed by the similarity in diffraction peak ratio observed across the bare copper and ALD-modified copper substrates (Supplementary Fig. 11b). In addition, the diffraction patterns after 0.5 mAh/cm$^2$ of lithium deposited on all substrates reveals similar lithium diffraction peaks, indicating that the preferred orientation of lithium does not vary across the substrates used in this study (Supplementary Fig. 11c).

To investigate the electrical properties, we measure the electrical resistivity for each type of substrate using a four-point probe. Electrical resistivity has a significant effect on lithium nucleation because nucleation is reliant on the supply of electrons for the reduction of lithium ions. We measure the resistivity of 50 nm of each metal oxide film used in this study, grown on Si wafers. We used 50 nm films to ensure reproducibility across film domains in our experiments. We find that the resistance to electron transport across films does not vary widely between ZnO and SnO$_2$ modified substrates (Supplementary Fig. 12). However,

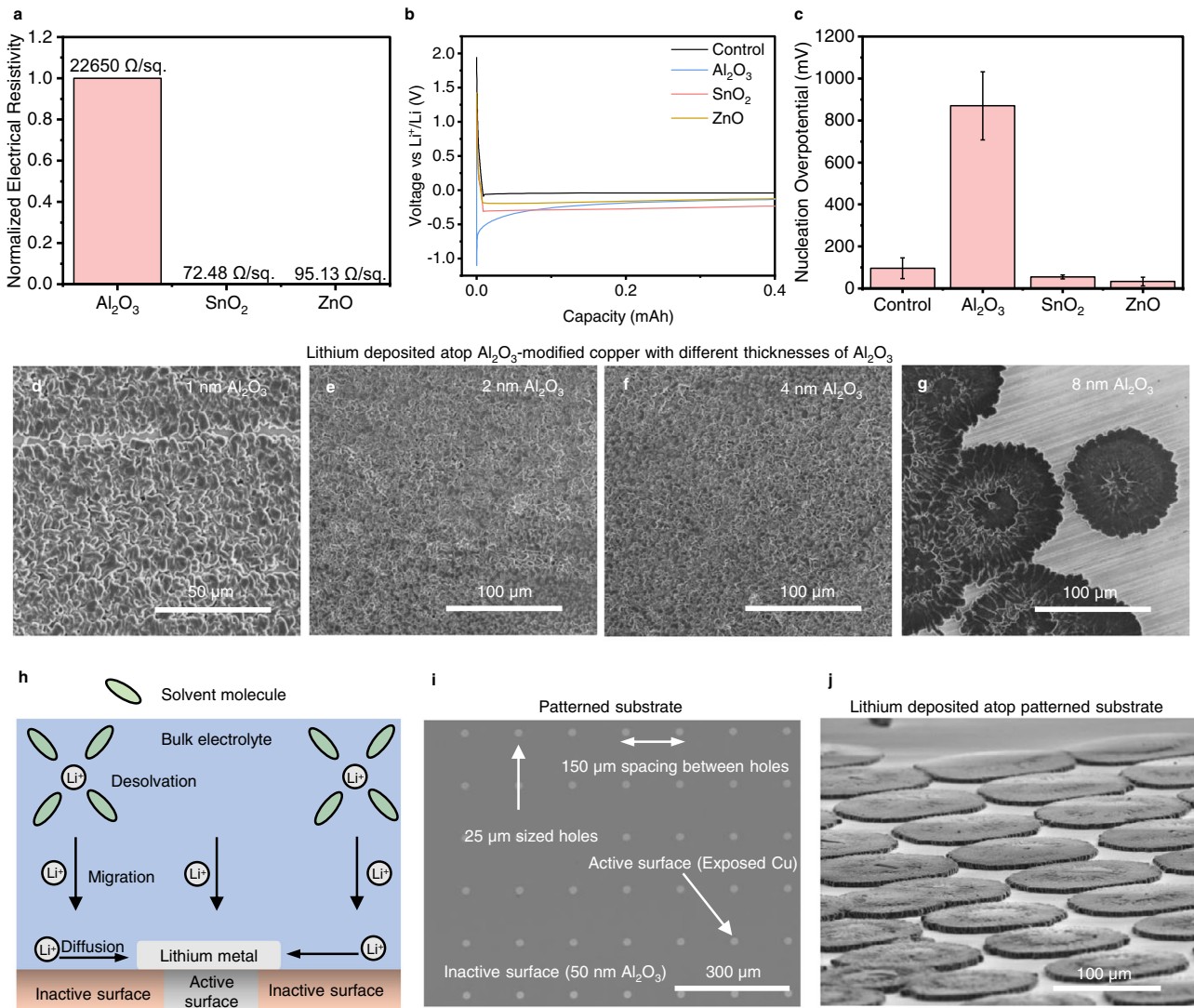

**Fig. 5 Understanding the intrinsic differences between bare copper and copper modified with ~8 nm of ALD films. a** Normalized electrical resistivity of ALD-modified substrates. **b** Voltage profiles showing the first cycle of lithium deposition at 1 mA/cm² and 1 mAh/cm² (with the *x*-axis plotted up to 0.4 mAh/cm² to accentuate the voltage profiles) on bare copper and ALD-modified copper. **c** Extracted nucleation overpotential of 1 mAh/cm² lithium plated at 1 mA/cm² on bare and ALD-modified copper substrates averaged over three cells, with each error bar representing one standard deviation from the average. **d–g** Top-view SEM images of 1 mAh/cm² of lithium plated at 1 mA/cm² on 1, 2, 4, and 8 nm of Al₂O₃-modified copper, respectively. **h** Illustration of lithium metal deposition on a resistive substrate. **i** Optical image of a 50 nm Al₂O₃-modified copper substrate with 25 μm-sized holes that expose the underlying copper substrate. **j** SEM image (~30° tilt) of 0.5 mAh/cm² of lithium deposited at 1 mA/cm² atop the patterned substrate shown in panel i.

the measured sheet resistivity for a similar thickness of Al₂O₃ is several orders of magnitude higher than those of ZnO and SnO₂, with values of 22,650, 95.13, and 72.48 Ω/square, for Al₂O₃, ZnO, and SnO₂, respectively (Fig. 5a). We provide details of the resistivity calculations in Supplementary Note 1. The high electrical resistivity of Al₂O₃ indicates that it strongly resists electron transfer during lithium electrodeposition, possibly explaining the sparsity of lithium deposits atop Al₂O₃.

To further explore this result, we deposit 1 mAh/cm² of lithium at 1 mA/cm² on all substrates: bare copper, and copper modified with 8 nm of Al₂O₃, 7 nm of ZnO, and 7 nm of SnO₂. Voltage profiles, with lithium deposition capacity plotted only up to 0.4 mAh to accentuate the voltage inflection point, show that the average nucleation overpotentials of lithium on bare copper, ZnO, and SnO₂ are 96 mV (±49 mV), 33 mV (±21 mV), and 54 mV (±9 mV), respectively, whereas the nucleation overpotential on Al₂O₃ is much higher at 870 mV (±162 mV) (Fig. 5b, c). Here,

each reported nucleation overpotential is an average value of at least three cells, with one standard deviation of that average reported in parentheses. These reported average first cycle nucleation overpotential trends were obtained using three cells for each substrate as shown in Supplementary Fig. 13, and subsequent cycles reveal that the overpotential trends are maintained (Supplementary Fig. 14). The correlation between the high nucleation overpotential and resistivity of Al₂O₃ suggests that Al₂O₃ resists the transport of electrons across its domains during the reduction of lithium ions, thus requiring a high overpotential for Li nucleation. As a result of this resistance to electron transport, the number of sites available for lithium nucleation are likely limited to the regions atop which electrons can transport across the film, for example at defects sites in the Al₂O₃ coating. This result suggests that the sparse particle density of lithium on Al₂O₃, as observed in Fig. 3d, is caused by a low number of available sites for lithium nucleation. Using a smaller lithium

capacity (0.05 mAh/cm$^2$) closer to the nucleation regime, we also observe much larger lithium deposits (~10 times in diameter) atop Al$_2$O$_3$ substrates compared to SnO$_2$ and ZnO substrates (Supplementary Fig. 15). The larger lithium deposits observed on Al$_2$O$_3$ after the onset of nucleation further support the argument that the reduction of electrical conductivity in Al$_2$O$_3$ reduces the number of nucleation sites, limiting the sites of Li growth to a smaller number of existing lithium nuclei.

We perform two additional tests to differentiate between chemical and electrical effects of the substrate as the origin of the unique, low surface area morphology of lithium. In the first study, we vary the thickness of Al$_2$O$_3$ (from 1 to 8 nm) deposited atop copper. While the four thicknesses studied should have the same chemical properties, their electrical resistance will increase with thickness. For the current collectors modified with 1, 2, and 4 nm of Al$_2$O$_3$, lithium deposits form uniformly atop all sites on the current collector (Fig. 5d–f and Supplementary Fig. 16). However, the lithium deposition morphology atop 8 nm of Al$_2$O$_3$ is drastically different, showing clustered deposits with preferential nucleation spots (Fig. 5g and Supplementary Fig. 16). The nucleation overpotential of lithium also differs significantly for different thicknesses of Al$_2$O$_3$, with values of 40, 40, 110, and 800 mV, for 1, 2, 4, and 8 nm of Al$_2$O$_3$, respectively (Supplementary Fig. 17). Drawing from the positive correlation observed between nucleation overpotential and resistivity shown in Fig. 5a–c, we can relate the behavior to electrical resistance. In the comparison of different Al$_2$O$_3$ film thickness, the 8 nm Al$_2$O$_3$ film has the highest resistance, resulting in its sparse lithium morphology. The disparity in lithium plating morphology atop Al$_2$O$_3$ films with varying thicknesses hence validates that electrical resistance plays a key role in controlling the morphology of lithium.

In a second experiment to confirm the role of electrical resistance at the current collector interface, we test an organic–inorganic hybrid material, hafnium–ethylene glycol (HfEG). HfEG has a very different chemical composition from Al$_2$O$_3$, but like Al$_2$O$_3$, films with Hf–O bonds are expected to have high electrical resistance[42]. Studies of lithium deposition atop a 6 nm film of HfEG (Supplementary Fig. 18) reveals sparse morphologies of lithium, similar to those observed on Al$_2$O$_3$. Moreover, the nucleation overpotential of lithium atop this HfEG film is 400 mV (Supplementary Fig. 19), suggesting that it is resistive as well, supporting the role of resistance in controlling lithium morphology.

Our results suggest that, atop resistive substrates, the likelihood for electron transport from the external circuit is reduced and possibly restricted to pinholes and defects, thereby limiting the nucleation of lithium to the few defect sites on the substrate. We propose that lithium deposition atop resistive substrates proceeds according to the model illustrated in Fig. 5h. The defect sites and pinholes in the resistive substrates represent the active sites while the other parts of the substrate, in which electron transport is prohibitive, are classified as inactive sites (Fig. 5h). After the desolvation of lithium ions from solvent molecules, lithium ions will migrate towards the surface of the current collector; however, nucleation will only occur at the active surface sites. Subsequently, lithium ions that impinge on the inactive surfaces of the current collector diffuse towards the active sites, to access electrons via lithium metal that is nucleated at the active surface sites (Fig. 5h).

This diffusion driven growth of lithium at steady state is fundamentally similar to the analytical treatment of diffusion-controlled currents at electrodes surfaces that contain electrically active and inactive areas[43,44]. Under short-time scales, it is derived analytically that each electrically active spot generates a linear diffusion field of species from the solution phase (perpendicular approach of species towards the electrode

surface)[43,44]. At longer time scales, the active surfaces become occluded by a diffusion layer, and as a result, diffusion becomes dominated by radial transport (non-perpendicular approach) of species via the inactive surface, towards the active electrode surface[43,44]. In a system like ours, where the electrically active surfaces are hypothesized to be small film defects, those surfaces behave like ultramicroelectrodes (UMEs), in which radial diffusion of species towards the active surfaces dominates linear diffusion even at very short time scales. A mathematical justification for this behavior is presented in Supplementary Note 2. This preference for lateral growth over vertical growth on resistive substrates is demonstrated using finite element simulations in Supplementary Fig. 20. We propose that the nucleation of lithium at the few active sites (defects) and the radial growth of lithium via diffusion of lithium ions from the inactive surfaces on the current collector are responsible for the sparse and planar morphology of lithium observed atop resistive substrates.

To verify this hypothesis, we deposit 50 nm of Al$_2$O$_3$ atop a copper substrate. Such a high thickness of Al$_2$O$_3$ is expected to reduce the likelihood of forming pinholes in the film[45]. Subsequently, we introduce 25 µm-sized holes, spaced 150 µm apart, atop the substrate to expose the underlying Cu substrate (Fig. 5i). This architecture mimics the model presented in Fig. 5h, with the exposed Cu surface representing the active surface and the remaining parts of the 50 nm Al$_2$O$_3$-modified substrate representing inactive surfaces. By depositing lithium atop the patterned substrate in Fig. 5i, we observe lithium morphologies that originate from the active surfaces and grow radially outward into flat, planar, pancake-like deposits (Fig. 5j and Supplementary Fig. 21). By observing the morphologies closely in Supplementary Fig. 21, it is evident that the particles formed on bare Cu are significantly smaller than the lithium deposits that grow atop Al$_2$O$_3$, supporting the diffusion model. We also observe the same lithium morphology atop 50 nm of Al$_2$O$_3$ patterned with 50 µm-sized holes (Supplementary Fig. 22). These results confirm that nucleation of lithium at resistive substrates occurs via defect sites (active surfaces) where radial growth of lithium is promoted through the diffusion of lithium ions from adjacent inactive surfaces.

The results clearly show that Al$_2$O$_3$-modified copper supports planar lithium morphology and improved cyclability in DDN (ether-based) electrolyte. We demonstrate the generalizability of Al$_2$O$_3$-modified copper through electrochemical tests in Li||Cu cells by using two other distinct classes of electrolytes—40 µL of 1 M LiPF$_6$ in EC/DEC with 10% FEC (EC/DEC/FEC) and 40 µL of 1 M LiFSI in FDMB (FDMB). The carbonate electrolyte (EC/DEC/FEC) used here is unstable in the presence of lithium metal, while the fluorinated ether (FDMB) is a state-of-the-art electrolyte for lithium metal[8], providing two extremes on a performance basis. Using Li||Cu cells, we cycle these cells under battery-relevant conditions for the respective electrolytes: 2 mA/cm$^2$ current density for the carbonate electrolyte and 1 mA/cm$^2$ current density for the fluorinated-ether electrolyte with a capacity of 1 mAh/cm$^2$ in each case. In both cases, the Al$_2$O$_3$-modified substrate improves cell performance over bare copper (Fig. 6a, b). In the FDMB-based electrolyte, cell lifetime is improved from 100 to 200 cycles (Fig. 6a), and in the carbonate-based electrolyte, cell lifetime is improved from 20 cycles to 80 cycles (Fig. 6b). These electrochemical improvements indicate that the Al$_2$O$_3$-modified substrate promotes reversible deposition and stripping of lithium in different classes of electrolytes.

We carry out SEM on the cells to identify the reason for performance improvements in the presence of alumina. Here, we freshly deposit 0.5 mAh/cm$^2$ of Li at 2 mA/cm$^2$ to investigate the growth morphology of Li. In FDMB, the lithium deposits on bare copper appear uniformly dispersed atop the current collector

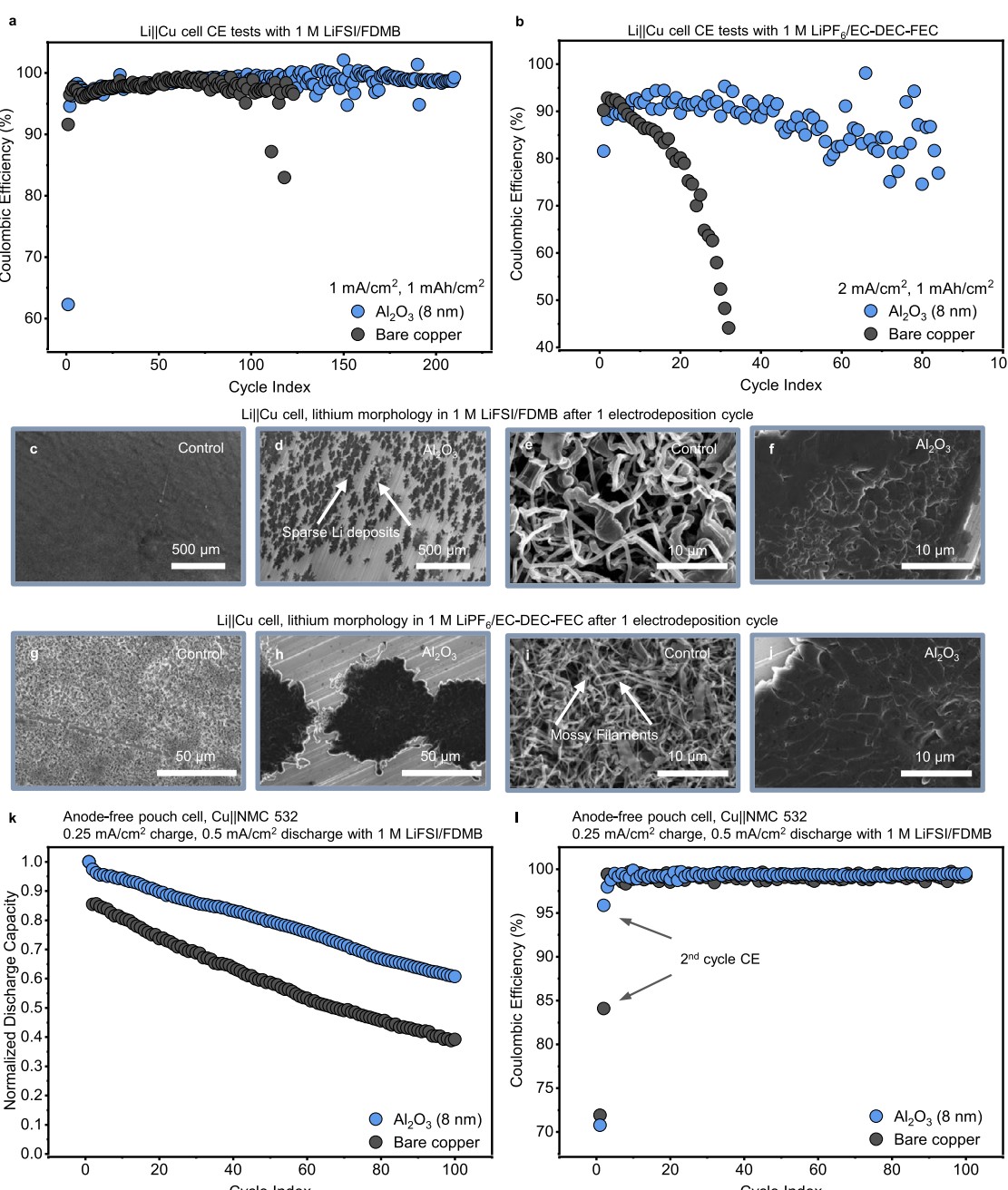

**Fig. 6 Al$_2$O$_3$-modified copper substrates improve performance in different classes of electrolytes. a** CE of Li||Cu cells cycled at 1 mA/cm$^2$ in FDMB electrolyte using bare copper and Al$_2$O$_3$-modified copper. **b** CE of Li||Cu cells cycled at 2 mA/cm$^2$ in EC/DEC/FEC electrolyte using bare copper and Al$_2$O$_3$-modified copper. **c–f** Top-view SEM images of 0.5 mAh/cm$^2$ of lithium plated at 2 mA/cm$^2$ in FDMB electrolyte atop bare copper and Al$_2$O$_3$-modified copper at two different magnifications. **g–j** Top-view SEM images of 0.5 mAh/cm$^2$ of lithium plated at 2 mA/cm$^2$ in EC/DEC/FEC electrolyte atop bare copper and Al$_2$O$_3$-modified copper at two different magnifications. **k** Normalized discharge capacity of anode-free Cu||NMC 532 pouch cells cycled at 0.25 mA/cm$^2$ (charge) and 0.5 mA/cm$^2$ (discharge) over the course of 100 cycles, with initial discharge capacities of 19.93 and 14.72 mAh for the bare Cu and Al$_2$O$_3$-modified Cu respectively. **l** Coulombic efficiency of cells shown in panel k.

(Fig. 6c). In contrast, the lithium deposits atop the alumina-modified substrate appear sparsely distributed (Fig. 6d), similar to what we observe in DDN electrolyte. In higher magnification images of the lithium deposits, they appear detached, filamentary, and isolated in the bare copper cell and coalesced and uniform in the Al$_2$O$_3$-modified cell (Fig. 6e, f). The superior uniformity of lithium deposits in the Al$_2$O$_3$-modified cell in the highly optimized FDMB electrolyte likely explains the improvement in performance demonstrated in Fig. 6a. In EC/DEC/FEC, the lower resolution SEM images reveal that in the bare copper cell,

deposits are formed atop every available exposed facet of copper while the deposits formed on the Al$_2$O$_3$-modified cells appear sparse, as observed in both the DDN and FDMB electrolytes (Fig. 6g, h). At higher resolution, the lithium deposits appear filamentary atop the unmodified copper (Fig. 6i). However, the lithium deposits once more appear uniformly coalesced on Al$_2$O$_3$-modified copper (Fig. 6j). These uniformly fused lithium deposits again are consistent with the electrochemical cycling improvements of Al$_2$O$_3$-modified copper over bare copper, since electrolyte consumption will be significantly reduced per cycle

and fewer deposits will break away to become inactive during cycling. In addition, the electrolyte-agnostic morphology of lithium atop $Al_2O_3$-modified copper reveals that its resistive properties could be a universal recipe for dendrite control. It is worth noting that even though resistive films outperform conductive films at lower current densities, the overpotential penalties associated with resistive films could limit their application at higher current densities.

We also extend these tests to practical anode-free pouch cells, using 2.5 mAh/cm$^2$ $LiNi_{0.5}Mn_{0.3}Co_{0.2}O_2$ (NMC 532) cathodes, cycled using 200 µL of 1 M LiFSI in FDMB (FDMB). The anode-free cells were cycled at 0.25 mA/cm$^2$ (17.86 mA/g NMC 532) charge and 0.5 mA/cm$^2$ (35.72 mA/g NMC 532) discharge, with the faster discharge rate chosen in accordance with a recent report which shows that faster discharge improves performance in anode-free cells[46]. After 100 cycles, the cell modified with 8 nm of $Al_2O_3$ retains 60% of its initial discharge capacity of 14.72 mAh while the cell with bare copper retains only 40% of its initial discharge capacity of 19.93 mAh (Fig. 6k). The $Al_2O_3$-modified cell outperforms the bare copper cell because it attains a Coulombic efficiency of 95.87% in the 2nd cycle while the cell with bare copper only attains a Coulombic efficiency of 84.08% in the 2nd cycle (Fig. 6l). In addition, the $Al_2O_3$-modified cell maintains a high discharge capacity over extended cycles with a 12.2% and 21.6% improvement in normalized discharge capacity over the bare copper cell in the 2nd and 100th cycles respectively, showing the high cycling reversibility of $Al_2O_3$-modified cells. We also test anode-free coin cells with 4 mAh/cm$^2$ $LiNi_{0.8}Mn_{0.1}Co_{0.1}O_2$ (NMC 811) cathodes, cycled under lean electrolyte conditions (5 µL) at 0.25 C (1 mA/cm$^2$) during charge and discharge, using EC/DEC/FEC (Supplementary Fig. 23). We find that cells with $Al_2O_3$-modified substrates outperform bare copper cells, reaching 25 cycles before losing 50% of their initial discharge capacity. Our finding introduces a new strategy for morphology control and fundamental insights into the benefits of lithium nucleation control that have the potential for improving lithium metal battery performance.

In conclusion, we present electrical resistance at the current collector surface as an electrolyte-independent strategy for controlling lithium metal morphology and improving lithium metal battery performance. By depositing thin films of $Al_2O_3$, $SnO_2$, and ZnO on copper, we report changes in lithium morphology that are dependent on the electrical resistance of the modified copper substrates. Low resistance substrates like copper and copper modified with $SnO_2$ and ZnO promote the formation of high surface area lithium deposits, whereas a high resistance $Al_2O_3$-modified substrate supports the formation of low surface area lithium deposits. We propose that the $Al_2O_3$-modified copper substrate reduces the available lithium nucleation sites by imposing a high charge transfer barrier, resulting in sparse nucleation only at defect sites. We demonstrate an unusual planar morphology of lithium observed atop $Al_2O_3$ that is likely caused by radial diffusion of lithium ions which diffuse laterally via non-defect sites towards the lithium deposits formed at defect sites. We report anode-free pouch cells, cycled under minimal external pressure, that maintain 60% of their initial discharge capacity after 100 cycles using $Al_2O_3$-modified substrates. Furthermore, we show that the $Al_2O_3$-modified copper induces similar low surface area lithium morphology across three distinct classes of electrolytes, and in each electrolyte, it displays significant performance improvement over low resistance substrates. In comparison to bare copper substrates in Li||Cu cells, we report ten-fold improvement in cycle life with DDN electrolyte, two-fold improvement with FDMB electrolyte, and four-fold improvement with EC/DEC/FEC electrolyte. This study presents a new parameter for

tuning lithium morphology and improving battery performance and poses important questions about the mechanisms behind existing lithium metal passivation strategies.

## Methods

**Film deposition.** For ALD $Al_2O_3$ deposition, trimethylaluminum (TMA) was used as the metal-organic precursor and water ($H_2O$) was the counter reactant. An ALD scheme of 1/30/1/30 s TMA pulse/purge/$H_2O$ pulse/purge sequence at 120 °C was adopted, which resulted in a growth rate of 1.1 per cycle.

For ALD $SnO_2$ deposition, tetrakis(dimethylamino) tin (IV) (TDMASn), heated to 60 °C, was used as the metal-organic precursor and water was the counter reactant. An ALD scheme of 1.5/5/1.5/5 s TDMASn pulse/purge/$H_2O$ pulse/purge sequence at 100 °C was adopted, which resulted in a growth rate of 1.8 per cycle.

For ALD ZnO deposition, diethyl zinc (DEZ) was used as the metal-organic precursor and water was the counter reactant. An ALD process involving a 0.1/5/1/5 s DEZ pulse/purge/$H_2O$ pulse/purge sequence at 120 °C was adopted, which resulted in a growth rate of 1.4 per cycle.

For the deposition of HfEG, tetrakis(dimethylamido) hafnium (TDMAH) heated to 60 °C was used as the metal-organic precursor and ethylene-glycol (EG) was the counter reactant. A deposition process involving a 1/20/120/1/20/120 s TDMAH pulse/soak/purge/EG pulse/soak/purge sequence at 120 °C was adopted, which resulted in a growth rate of 1.9 per cycle. During soak, the pump and $N_2$ flow are both turned off to improve growth kinetics.

All depositions were conducted directly on Cu foil in a Gemstar 6 ALD reactor (Arradiance). The film thicknesses were determined by growth on a reference Si wafer, using a J.A Woollam M2000 Variable Angle Spectroscopic Ellipsometer at 65° and 70° angles of incidence and wavelengths ranging from 210 to 1688 nm.

**Materials.** All electrolytes were prepared and handled in an Ar-filled glove box in which $O_2$ concentration was below 0.2 ppm and $H_2O$ concentration was below 0.01 ppm. The DDN electrolyte was prepared using 1 M LiTFSI (Solvay), a 1:1 mixture of DME and DOL (both purchased from Aldrich), and 1% by weight $LiNO_3$ (Aldrich). The EC/DEC/FEC electrolyte was prepared using 1 M $LiPF_6$ in EC:DEC received from Gotion and 10% by volume FEC (BASF). The FDMB solvent was synthesized as follows: Using a 1000 ml round-bottom flask, 400 ml of anhydrous tetrahydrofuran and 64 g of 2,2,3,3-tetrafluoro-1,4-butanediol were mixed, cooled to 0 °C and stirred for 10 min. Subsequently, 40 g of NaH (60% in mineral oil) was added in batches and stirred at 0 °C for 30 min. Afterwards, 140 g of methyl iodide was added slowly to the stirring suspension and the ice bath was removed to enable the suspension warm up to room temperature. After stirring for 2 h at room temperature, the round-bottom flask was slowly heated to 60 °C to reflux overnight. After reaction completion, the flask was cooled down to room temperature, the mixture was filtered, and solvents were removed under vacuum. The crude product was distilled under vacuum (~45 °C under 1 kPa) three times to yield a final, colorless liquid product. The FDMB electrolyte was prepared using 1 M LiFSI (Oakwood) in the FDMB solvent. High-purity Li foil (0.75 mm, 99.9% Alfa Aesar), Cu foil (Pred Materials), polymer separator (Celgard 2325), NMC 532 (Targray), NMC 811 (Targray) were used to make cells in different configurations, with their specific combinations detailed below.

**Electrochemistry.** Type 2032 coin-cells were assembled in an argon glovebox with a polymer separator (Celgard 2325). Li metal foil (0.75 mm thick, 99.9% Alfa Aesar) was used as the counter/reference electrode and a Cu foil was used as the working electrode. Li foils were punched to 1 cm$^2$. Li was mechanically sheared to remove surface oxides and improve electrical contact while Cu foil was rinsed with acetone, isopropyl alcohol, and deionized water to remove surface contaminants prior to cell assembly. This Li||Cu configuration was used for cyclability, CV measurements, SEM, and XPS characterizations. Li||Cu cyclability tests in coin-cells were carried out with a Li dissolution cutoff voltage of 1 V versus Li/Li$^+$ and the corresponding electrodeposition cutoff capacities stated in the text using an Arbin battery cycler. Nucleation overpotential in Li||Cu cyclability tests was calculated as the difference between voltage at the inflection point and the lithium growth region (flat region) of the voltage curve during lithium deposition. Li||Cu CV measurements were carried out within a voltage window of 0.015–3 V versus Li/Li$^+$ at a scan rate of 1 mV/s using a Biologic VMP3. Anode-free coin-cells were prepared using Soem 2032-type coin cells with 1 cm$^2$ Cu foil (Pred Materials) or 1 cm$^2$ ALD-modified Cu foils, 1 cm$^2$ NMC 811 (4 mAh/cm$^2$, Targray), Celgard 2325, and 5 µL of EC/DEC/FEC. These anode-free Cu||NMC 811 cells were cycled at ~25 °C within a voltage window of 3.0–4.3 V versus Li/Li$^+$. Anode-free pouch cells were assembled using Cu foil (Pred Materials) or ALD-modified Cu foils, NMC 532 (2.5 mAh/cm$^2$ loading, Targray), Celgard 2325 as the separator, Al-plastic as packaging, and 200 µL of FDMB electrolyte. The anode-free Cu||NMC 532 pouch cells were cycled within a voltage window of 3.0–4.4 V versus Li/Li$^+$. The pouch cells were cycled at ~25 °C and clamped using woodworking vises at an approximate pressure of 1000 kPa.

**Microscopy.** All samples were rinsed with the corresponding pure solvents (diethyl carbonate for carbonate-based electrolytes, and 1,2 dimethoxy ethane for ether-

based electrolytes) and dried inside the Ar glovebox before microscopy. For cryo-TEM analysis, samples were plunge frozen in liquid nitrogen without air exposure, in accordance with previous reports[35,47]. Samples were loaded onto a Gatan 626 cryogenic TEM holder under liquid nitrogen and maintained at −175 °C within the TEM column. Cryo-TEM measurements were carried out using a FEI Titan 80–300 environmental (scanning) TEM operated at an accelerating voltage of 300 kV. The instrument was equipped with an aberration corrector in the objective lens which was tuned before each sample analysis. Scanning electron microscopy was performed using a FEI Magellan 400 XHR. Optical microscope images were collected using an Olympus BX-51.

**XPS characterization**. Cu foil working electrodes with Li freshly deposited using DDN electrolyte were prepared in an Ar glove box and rinsed with 90 μL of DME to remove residual Li salts, then transferred to an XPS chamber using a vacuum transfer vessel. XPS signals were collected on a PHI VersaProbe 1 scanning XPS microprobe with an Al Kα source.

**X-ray diffraction**. X-ray diffraction measurements on bare copper and ALD-modified copper were performed using a PANalytical X'Pert PRO X-ray diffraction system with Cu Kα radiation and an X-ray tube working power of 45 kV/40 mA. X-ray diffraction measurements on lithium were performed using a PANalytical Empyrean with Mo Kα radiation. To prevent air exposure, lithium samples were sealed in an air-free vessel under Ar in the glovebox before diffraction experiments.

**Finite-element simulations**. We develop a numerical model to understand the electrochemical behavior and morphological evolution of Li nucleation during continuous Li plating. Two models were built to represent lithium nucleation on conductive and resistive substrates. In the model for conductive substrates, each lithium particle is represented with a diameter of 5 μm, spaced 2.5 μm between each other. In the model for resistive substrates, each lithium particle is represented with a diameter of 5 μm, spaced 200 μm between each other. The Li ion transport in electrolyte and the charge transfer reactions at the Li/electrolyte interface are described by the Nernst–Planck equation and the Butler–Volmer equation, respectively, and numerically solved by the electrochemical module integrated in COMSOL Multiphysics software. The deformation of Li electrodes caused by Li deposition is simulated by the deformed geometry module, which is coupled with the electrochemical module in COMSOL. Details of the electrochemical model are included in the Supplementary Information. The geometrical and electrochemical parameters in the numerical model are set to be consistent with the experimental setup. The galvanostatic Li plating is simulated by applying a constant current density (1 mA/cm$^2$) on the Li metal electrode. The physical properties of Li metal and the electrolyte are listed in Supplementary Table 2.

The fluxes of the Li ions in the electrolyte are described as

$$\frac{\partial C_l}{\partial t} + \nabla \cdot \mathbf{J}_l = 0 \tag{1}$$

$$\mathbf{J}_l = -D_l \nabla C_l + \frac{\mathbf{i}_l t_+}{F} \tag{2}$$

where $\mathbf{J}_l$ represents Li$^+$ flux in the electrolyte/active material, $D_l$ represents the Li$^+$ diffusivity in the electrolyte, $C_l$ is the concentration of Li$^+$ in the electrolyte, $t_+$ is the transference number of Li$^+$, $F$ is the Faraday's constant, and $\mathbf{i}_l$ is the electric current density in the electrolyte, which is governed by the migration and diffusion of Li$^+$,

$$\mathbf{i}_l = (-K_l \nabla \phi_l) + \frac{2 K_l R T}{F}\left(1 + \frac{\partial \ln f}{\partial \ln C_l}\right)(1 - t_+) \nabla \ln C_l \tag{3}$$

where $K_l$ is the electrolyte conductivity, $\phi_l$ is the electric potential in the electrolyte, $R$ is the gas constant, $T$ is the temperature, and $f$ is the mean molar activity coefficient of the electrolyte.

At the Li/electrolyte interface, the charge transfer reaction is described by the Bulter–Vomer equation

$$i = i_0 \left(\exp\left(\frac{\alpha_a F \eta}{RT}\right) - \exp\left(-\frac{\alpha_c F \eta}{RT}\right)\right) \tag{4}$$

where $\alpha_a$ ($\alpha_c$) is the anodic (cathodic) transfer coefficient, $\eta$ is the overpotential, and $i_0$ is the exchange current density. The overpotential $\eta$ of an electrochemical reaction is defined as,

$$\eta = \phi_s - \phi_l - E_{eq} \tag{5}$$

where $E_{eq}$ is the equilibrium potential for the electrochemical reaction, and $\phi_s$ is electric potential of the electrode. The exchange current density $i_0$ is defined as

$$i_0 = i_{0ref}\left(\frac{C_l}{C_{lref}}\right)^{\alpha_a} \tag{6}$$

where $i_{0\_ref}$ is the reference exchange current density while $C_{l\_ref}$ is the electrolyte reference concentration.

**Film patterning experiments**. To prepare the Al$_2$O$_3$-modified current collector (CC) for patterning, the 50 nm Al$_2$O$_3$ sample was cleaned with acetone and isopropyl alcohol, dried with pressurized nitrogen, then mounted on the center of a 4-inch wafer with Kapton tape. The sample was passed through a Silicon Valley Group (SVG) coater and coated with 1.6 μm of SPR 3612 positive resist. Subsequently, a soft bake was carried out at 125 °C for 60 s, then the sample was patterned on the Heidelberg MLA 150 using direct write lithography. Each hole on the pattern was created by exposing an area to 375 nm ultraviolet rays. After lithography, the exposed resist was developed on the SVG developer. A post-exposure bake was carried out at 110 °C for 90 s, after which an MF-26A developer was dispensed on the sample, followed by a DI water rinse. A hardbake was carried out at 110 °C for 120 s to evaporate the residual water.

The patterned CC was then transferred to a wet bench for hydrofluoric acid (HF) etch. 100 μL of 1 vol% HF was deposited atop each patterned spot on the CC for 90 s. After 90 s of etching, a pipet was used to remove the HF solution, then the surface of the etched region was rinsed with deionized water. The SPR 3612 resist was removed with acetone and isopropyl alcohol to reveal the patterned substrate.

## Data availability
The relevant datasets generated and analyzed in this study are provided with this paper. Source data are provided with this paper.

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

## Acknowledgements

S.T.O. acknowledges support from the Knight-Hennessy scholarship for graduate studies at Stanford University. Y.C. acknowledges support from the Assistant Secretary for Energy Efficiency and Renewable Energy, Office of Vehicle Technologies, of the U.S. Department of Energy under the Battery Materials Research (BMR) Program and the Battery500 Consortium program. Part of this work was performed at the Stanford Nano Shared Facilities (SNSF), supported by the National Science Foundation under award ECCS-1542152. The authors thank Yinxing Ma for help with XRD experiments on lithium metal.

## Author contributions

S.T.O., Y.C. and S.F.B. conceived the idea and designed the experiments. S.T.O. performed ALD, XPS, four-point probe experiments, and all electrochemical experiments. W.Z. and Y.Ya. performed SEM. W.Z. and A.S. carried out cryo-TEM and patterning experiments. R.X. performed finite element COMSOL simulations. J.A.R. performed XRD on ALD-modified Cu. Z.Y. from Zhenan Bao's group synthesized and provided the FDMB electrolyte. W.H., Y.Ye. and D.T.B. assisted with electrochemical experiments. J.R.S. helped with ALD and provided helpful discussions. S.T.O. wrote the first draft of the manuscript and W.Z., W.H., D.T.B., Y.C. and S.F.B. edited and revised the manuscript. All authors read and discussed the manuscript.

## Competing interests

The authors declare no competing interests.
