## [Peer Review File · Nature Communications]

Reviewer comments, first round review

Reviewer #1 (Remarks to the Author):

The study of lithium electrodeposition presented in the manuscript is timely and important to realising higher-energy batteries, which are essential for the transition from fossil fuel-based to electric vehicles.

The presented research is comprehensive and of interest to the Electrochemistry community. Although using ALD coatings as artificial SEIs (or nucleation layers) is not new as a concept, the authors present a comprehensive comparison between three known coatings, which are of significance to the field of anode-free lithium metal batteries.

The work meets the expected standards in the field, however, in several parts of the manuscript, the authors haven't provided sufficient data points or information regarding the statistical significance of the measurements. The authors suggest an alternative analysis of the results, which is often in contrast to other studies in the literature; however, these discrepancies are not discussed. Currently, there isn't enough detail provided in the methods for the work to be reproduced. Thus, I recommend accepting the manuscript after major revision.

Comments:

1. Page 3, row 47: since electrolyte engineering is done before cell assembly, it usually doesn't qualify as an in-situ method. If the authors suggest otherwise, a discussion should follow.
2. Page 4, row 89 and throughout the manuscript: the authors suggest that lithium morphology is not affected by the SEI; however, it is well established in the literature that the SEI properties are determined by the electrode (specifically the metal). For example, SnO₂ is a well-known lithium alloy anode with a distinctive SEI composition and structure. A discussion of this discrepancy is essential.
3. Although the authors suggest that the studied layers serve as nucleation layers, lithium oxide (Li₂O), a known SEI component attributed to the reduction of copper oxide, appeared in all the SEIs (Supplementary Figure 3a). Moreover, the intensity of the Li₂O peak is higher for Al₂O₃ and ZnO coated anodes. Please discuss the possible reactions which result in Li₂O in the SEIs.
4. The authors measured the Coulombic efficiency of Li vs Cu half cells. In this type of cell, the counter electrode is an infinite lithium reservoir compared to the amount of the plated lithium. As a result, the calculated efficiency is the ratio between the applied charge and the recoverable charge in the following cycle. However, it doesn't reflect the lithium loss due to parasitic reactions and can't distinguish between lithium plating and other reversible electrochemical processes (such as tin or copper reduction). While the analysis complies with the definition of Coulombic efficiency, using it to predict the potential performance of a battery cell should be done with care and supported with other measurements. Coulombic efficiency lower than 99.5% is considered not practical due to the expected short cycle life.
5. A comment on the statistical significance of the findings should be added. The number of cycled coin cells and SEM samples should be indicated. Error bars should be added to the SEI thickness in Figure 4c, nucleation overpotential on Figures 5c, and Supplementary Figures 8,9. Error-values should be added to the average Li nuclei sizes (Figure 3) and SEI thickness (Figure 4).
6. The authors suggested that the lithium deposits on SnO₂ (Figure 3) are isolated from the current collector. However, from the SEM image, it is apparent that this morphology is interconnected better than the lithium nuclei plated on other substrates. Moreover, lithium morphologies on the bare copper and ZnO are very similar; while SnO₂ has a very different morphology, it seems that it

provides the best current collector coverage. Does the comparison of Li morphologies valid in this case? Was the morphology of a thinner Li layer on SnO₂ observed?

7. Figure 1 illustrates the different lithium electrodeposition mechanisms as a function of the coating. However, it is confusing since the SEI is not shown on either the active or inactive surfaces. Moreover, it should be emphasised that the SEI's composition and stability could vary between various substrates, especially while some of them alloy lithium.

8. Irreversible capacity:

a. How do the non-homogeneous Al₂O₃ coating on Li and Cu affect the SEI formation during the first and further cycles?

b. Could a larger exposed surface on the bare copper encourage larger irreversible capacity at the first cycle?

9. Page 7, row 125: could the capacity loss be attributed to dead lithium formation and galvanic corrosion in addition to the SEI formation? If SEI formation is suggested as a sole degradation method, further justification is needed.

10. LiNO₃ is used in all the studied systems. Does it react differently with the various surfaces explored in this manuscript?

11. Figure 2: the range of the Y axis should be minimised to 90-100%, and the size of the symbols should be decreased to emphasise the subtle difference between the efficiency of the cells based on the various coatings.

12. The XPS analysis is focused on the C-O and C-F bonds. However, a more detailed analysis of the XPS data is required; specifically, the peaks should be assigned either to the product of the salt or solvents' reduction products.

a. What was approximately the thickness of the sputtered layer in the XPS experiment?

b. Was a reference of the dried electrolyte or electrolyte pristine salt measured?

c. Additional XPS measurements are required since further sputtering must be done until the coating or copper signal appears. These additional experiments are essential to the comparison of the SEI composition between the different nucleation layers. Specifically, an XPS measurement and depth profile of the SEI formed on 8 nm Al₂O₃ coating is critical.

13. In Figures 3i-3k, the lighter contrasted features were assigned to "dead lithium". Could these features also be identified as a very thick SEI or non-reacted electrolyte layer?

14. Supplementary Figure 3b, a discussion of the S 2p spectra is missing.

15. Was the amount of "dead Li" measured or estimated?

16. Impedance and electronic conductivity measurements:

a. Please provide further experimental details on the impedance measurements.

b. How was the charge transfer impedance estimated? Please provide the equivalent circuit used for the fitting and the fitting parameters.

c. How many EIS experiments were conducted? What is the error for the charge transfer impedance? Are the differences in charge transfer impedance and electrical conductivity statistically significant?

d. What is the physical meaning of the solution resistance?

e. The axes of the Nyquist plots should be of the same size; otherwise, the shape of the spectra is distorted and loses physical meaning.

f. Please discuss the effect of the surface area of the deposited Li on the measured impedance. Were the ratios between the charge transfer impedance values validated using another method?

17. Page 12 row 216: could the Al signal be a result of Al corrosion?

18. Overpotential

a. The authors attributed the larger nucleation overpotential to higher electronic resistance of the substrate, in contrast with the common understanding in the literature, attributing larger

overpotential to more heterogeneous metal plating. Please contrast these findings with several studies in the literature.

b. Was a similar overpotential ratio observed in the following cycling of the half and full cells?

19. The authors demonstrated a mushroom-like Li growth on the Al₂O₃ coated substrates. Were any morphological or composition differences between the surface in the centre of the round Li deposits and on the edges found?

20. A coating of hafnium-ethylene glycol (HfEG) was briefly introduced. Was this type of coating used before in alkali metal batteries? How does this coating compare with the 8nm Al₂O₃ coating?

21. The performance of the 8 nm Al₂O₃ coating 8nm was utilised in Cu//NMC 532 pouch cells (Figure 6k and 6l). The scalability of the coating method is impressive; however, the slopes of the coated and bare copper-based cells are very similar, and it seems that the difference is caused merely by the capacity drop at the first cycle. Please discuss the various possible degradation mechanisms and the contribution of the Al₂O₃ coating in the experiment presented in Figure 6k.

Reviewer #2 (Remarks to the Author):

The authors present a comprehensive study of the effect of various passivation strategies on the Li deposit morphology and electrochemical performance. The authors show that these effects are also present when the electrolyte is varied. The work presented by Oyakhire and co-workers is well structured and clearly presented, while the results provide enough novelty and thoughtful discussion, therefore, the manuscript, is recommended for publication in Nature Communications. Below are some questions that can be considered by the authors to further be addressed in their manuscript:

1. The voltage profiles for initial cycles should be reported (maybe in SI), as this would clarify some of the questions raised and also the discussions made by the authors.

2. In Figure 3, what is preventing Li deposits to grow in length towards the CE on the copper modified with Al₂O₃? Please discuss.

3. What is the initial resistance of the investigated cells? Regarding the Figure S2, the higher amount of accumulated dead Li in the cell, would not result in a higher resistance of such cells? Table 1 can be further supplemented with such information.

4. The increase of the electrolyte resistance was assigned on the changes in the electrode morphology and the subsequent reaction with the electrolyte components. Can you subtract the effect of the electrode's composition and morphology from the cells cycled for 50 cycles, by measuring the electrolyte resistance with blocking electrodes? This would strengthen the statement at page 11, line 201.

5. Regarding the studies on deposited Li, did you investigate the morphology of the Li nucleates and their location, when a smaller amount of Li is deposited as compared to the 1mAh/cm²? Please discuss.

6. Regarding the XPS investigations, these investigations were carried out for the samples after 1mah/cm² was deposited. How is the outcome of the XPS measurements in accordance with the statement of the EIS measurements?

7. What is the effect of the chemical composition of the SEI on low amount of Li deposited (maybe 0.1 mAh/cm²) during an initial cycle? Then these XPS results would be complementary to the cryo-TEM investigations.

8. With respect to the statement at line 246, are there other effects that degrade the cell

performance with time for various substrates? What about the interface Cu|electrolyte? (where the Cu substrate contains Li deposits, thus this substrate would be at the potential of metallic Li or a mixed potential)

9. Regarding Figure 4d-f, in the case of lithiophilic coatings (i.e. SnO₂ and ZnO) the formation of Zn and Li₂O would occur prior the Li deposition. When further Li is deposited, LiZn will form and this will be also reversible. Have you carried out cyclic voltammetry to 15mV in order to further investigate if these reactions are not occurring? This should be carefully investigated, as also in Figure 5b there is an deflection (prior the inflection point and well above 0V) observed for Cu, ZnO@Cu and SnO₂@Cu as compared to Al₂O₃@Cu.

10. The authors state that such a behavior, as the one discussed at line 278-280, is due to the difference in kinetic affinity for Li towards electrolyte instead of lithiophilic layer. It is not clear what Li is expected at 15mV on the surface of Cu substrate. Please discuss.

11. What is the electrical resistivity of the 8nm ALD coatings on Copper that were investigated in this manuscript? Are then the results in accordance to those measured for 50nm thick coatings?

12. Regarding the existence of pin holes, have you tried to deposit Al₂O₃ at higher temperature than 120°C?

13. Would the effect of a surface modified 8nm Al₂O₃ Cu substrate be similar to that observed for the 50nm Al₂O₃ substrate, when Li will be deposited?

14. There are some early reports that state that Al₂O₃ prepared at low temperature and in a thin layer is also partially conductive for Li-ions. Is then, the formation of the Li deposits mainly related to the existence of defects in the coating?

15. Regarding the use of Li|Cu cells and Cu|NMC cells, could you compare the capacity retention that is recorded with these cells? From the comparison of only the Coulombic efficiency, one cannot get the whole picture of the negative electrode's performance. Even with the negative electrode's substrate modification, why is there a capacity loss observed in full cells?

Reviewer #3 (Remarks to the Author):

In this manuscript, the author deeply studied the changes of lithium deposition morphology after modifying copper collector by ALD deposition film, and explored the difference between conductive film and resistive film in the process of lithium deposition by using a variety of characterization methods. The Al₂O₃-modified cells show significantly higher cycling stability, reaching a cycle index of 390 at an average CE of 96 % before showing signs of capacity fade. Therefore, this manuscript is acceptable after addressing the following issues.

1) In Figure 5b-5c, the nucleation overpotential of lithium on ZnO and SnO₂ is larger than that on bare copper. ZnO and SnO₂ are conductive nanofilms, should have higher affinity for lithium deposition. Please explain why is the nucleation overpotential larger?

2) In Figure 6k, the difference in capacity retention between Al₂O₃-modified copper and bare copper is mainly in the first cycle. Does it mean that there is not much difference between bare copper and Al₂O₃-modified copper after the formation of the SEI on the bare copper?

3) Resistive nanofilm may play a role in low current, but can they be better than conductive nanofilms at higher current? It is hoped that the rate performance of Li/Cu cells using bare copper and copper modified with 7-8 nm of SnO₂, Al₂O₃, and ZnO can be compared.

4) What is the effect of the single defect size of the resistive nanofilm on the performance of Li/Cu or anode-free Cu || NMC 532 cells. Does the total surface area of all defects affect the performance?

5) The citation format of references should be consistent. For example, reference 11 has DOI number and should be modified.

Response to Reviewers for “Electrical Resistance of the Current Collector Controls Lithium Morphology”

We are very grateful to the reviewers for providing valuable feedback that improves our paper. We believe that the comments and questions raised by the reviewers have helped us clarify our arguments and show the reproducibility of our work. We respond to each question raised by the reviewers in blue, and highlight the changes made to our manuscript in yellow.

Reviewer #1 (Remarks to the Author):

The study of lithium electrodeposition presented in the manuscript is timely and important to realising higher-energy batteries, which are essential for the transition from fossil fuel-based to electric vehicles. The presented research is comprehensive and of interest to the Electrochemistry community. Although using ALD coatings as artificial SEIs (or nucleation layers) is not new as a concept, the authors present a comprehensive comparison between three known coatings, which are of significance to the field of anode-free lithium metal batteries.

The work meets the expected standards in the field, however, in several parts of the manuscript, the authors haven't provided sufficient data points or information regarding the statistical significance of the measurements. The authors suggest an alternative analysis of the results, which is often in contrast to other studies in the literature; however, these discrepancies are not discussed. Currently, there isn't enough detail provided in the methods for the work to be reproduced. Thus, I recommend accepting the manuscript after major revision.

We appreciate the reviewer's careful reading of our manuscript and have addressed their comments below. We have added more data to show the reproducibility of our work, and we believe that the edits improve the quality of our manuscript.

Comments:

1. Page 3, row 47: since electrolyte engineering is done before cell assembly, it usually doesn't qualify as an in-situ method. If the authors suggest otherwise, a discussion should follow.

Response: We thank the reviewer for pointing this out. While the process of electrochemically forming the SEI from electrolyte decomposition is often described in the literature as an in-situ method, we understand how electrolyte engineering could also be considered as an ex-situ method. As a result, we have removed the phrase “in-situ” from the main text. Here is the revised text:

Revised text on page 3, line 45: Efforts towards passivating lithium have been focused on the design of stable interphases between lithium and the electrolyte, many of which have been implemented using molecular techniques like electrolyte engineering¹⁻⁴ and thin film methods like atomic and molecular layer deposition.⁵⁻⁹

2. Page 4, row 89 and throughout the manuscript: the authors suggest that lithium morphology is not affected by the SEI; however, it is well established in the literature that the SEI properties are determined by the electrode (specifically the metal). For example, SnO₂ is a well-known lithium alloy anode with a distinctive SEI composition and structure. A discussion of this discrepancy is essential.

Response: Our statement regarding the similarity of SEIs across all our ALD films should not be interpreted as the fact that the SEI does not impact lithium morphology because, as the reviewer points out, that is not the case. Numerous studies have drawn correlations between the properties of the SEI and the corresponding lithium morphology.¹⁰⁻¹² Instead, what we are stating is that since the SEIs across all our ALD films are the same, then changes in lithium morphology must originate from the ALD substrates. Also, we agree with the notion that the decomposition of liquid electrolytes is dependent on electrode properties. However, in our study, since most of the electrolyte decomposition occurs atop lithium, the formation of similar SEIs across our ALD films is expected. We have modified our paper thus to clarify this.

Revised text on page 4, line 86: We show that the solid electrolyte interphase (SEI) formed from electrolyte decomposition atop lithium in the presence of each modified copper substrate is chemically and structurally similar, indicating that in this work the changes in lithium morphology stem from differences in substrate properties and not the SEI.

3. Although the authors suggest that the studied layers serve as nucleation layers, lithium oxide (Li₂O), a known SEI component attributed to the reduction of copper oxide, appeared in all the SEIs (Supplementary Figure 3a). Moreover, the intensity of the Li₂O peak is higher for Al₂O₃ and ZnO coated anodes. Please discuss the possible reactions which result in Li₂O in the SEIs.

Response: We appreciate the reviewer's comment regarding the Li₂O peak in our XPS figures. While the Li₂O could originate from reactions between lithium and CuO during SEI preformation,¹³ these reactions contribute minimally to the Li₂O peak in the SEI because of the absence of an SEI preformation step in our electrochemical protocol. Instead, we think that the Li₂O formed in our cells is mainly due to direct reactions between Li and the electrolyte. The formation of Li₂O from Li/electrolyte reactions in our electrolyte (DDN) is well documented in the literature.¹⁴ Also, the relative intensities of the Li₂O peaks compared across all substrates may not be too meaningful because XPS peak intensities vary across the surface of lithium during due to the heterogenous nature of the SEI.¹⁵ Instead, we believe that there is much more meaning in identifying the species across the SEI, to confirm that the reactions at the Li-electrolyte surface are the same across each substrate. We have added the following statement to our manuscript, to clarify this:

Revised text on page 13, line 243: It is noteworthy that we focused on identifying the chemical species in the SEI rather than quantifying them due to the variability in SEI composition across the surface of Li. Instead, we use the presence of similar SEI chemical composition across our substrates to show that they have similar Li-electrolyte interfaces.

4. The authors measured the Coulombic efficiency of Li vs Cu half cells. In this type of cell, the counter

electrode is an infinite lithium reservoir compared to the amount of the plated lithium. As a result, the calculated efficiency is the ratio between the applied charge and the recoverable charge in the following cycle. However, it doesn't reflect the lithium loss due to parasitic reactions and can't distinguish between lithium plating and other reversible electrochemical processes (such as tin or copper reduction). While the analysis complies with the definition of Coulombic efficiency, using it to predict the potential performance of a battery cell should be done with care and supported with other measurements. Coulombic efficiency lower than 99.5% is considered not practical due to the expected short cycle life.

Response: The reviewer makes an excellent point. We agree that Li/Cu cells are not perfectly representative of real battery performance; yet, under lean electrolyte conditions, they help identify how quickly electrolytes are consumed by lithium deposits.¹⁶ In our Li/Cu experiments, the main objective was to identify how quickly our electrolyte was consumed based on the cycle life of the Li/Cu cells. To display more realistic battery performance, we cycled anode-free cells under lean electrolyte conditions to show the benefits of using the ALD coatings for morphology control, as shown in *Figure 6* of our main text.

5. A comment on the statistical significance of the findings should be added. The number of cycled coin cells and SEM samples should be indicated. Error bars should be added to the SEI thickness in *Figure 4c*, nucleation overpotential on *Figures 5c*, and Supplementary *Figures 8,9*. Error-values should be added to the average Li nuclei sizes (*Figure 3*) and SEI thickness (*Figure 4*).

Response: We have added statements and calculations that reflect the repeatability of our results to the main text thus:

Revised text:

Page 7, line 126: Significantly, the Al₂O₃-modified cells show higher cycling stability, reaching a cycle index of 390 at an average CE of 96 % before showing signs of capacity fade. Cycling performance is verified using at least two cell replicates (Supplementary *Figure 2*).

Supplementary Figure 2. CE of replicate Li/Cu cells cycled at 1 mA/cm² using copper modified with 7-8 nm of a) SnO₂, b) ZnO, and c) Al₂O₃. Red and blue markers represent separate cell measurements.

Page 10, line 156: In comparison, the lithium morphology on SnO₂ has a long, snake-like structure that appears isolated from the current collector, with approximate diameter of 3.77 μm ($\pm 1.36 \mu\text{m}$) (*Figure 3b*). The tendency for lithium to coalesce into long deposits of this morphology could be an effect of the

interfacial energy between SnO₂ and lithium, and the apparent electrical isolation of the deposits may explain the quick decline in performance of the SnO₂-modified cells. Additional SEM images that show the isolation of Li particles atop SnO₂ are shown in Supplementary Figure 3.

Supplementary Figure 3. Li particles deposited on 7 nm of SnO₂ shown in a) High magnification , and b) Low magnification.

Page 10, line 166: In contrast to all the other substrates, Al₂O₃ supports sparse lithium nucleation, with a distinctive Li morphology of aggregated clusters of approximately 91.47 μm (± 43.9 μm) in diameter (Figure 3d). These aggregated lithium deposits on the Al₂O₃-modified copper substrate are expected to require reduced electrolyte consumption during cycling, an effect which would explain the large improvement in cycling reversibility of Al₂O₃-modified copper shown in Figure 2a. Low magnification SEM images show that the Li particles captured in Figures 3a-3d are representative (Supplementary Figure 4), and size distribution analysis obtained using at least 10 distinct Li particles is displayed in Supplementary Figure 5.

Supplementary Figure 4. SEM images revealing representative Li particles across the a) Bare Cu, b) SnO₂, c) ZnO, and d) Al₂O₃ substrates respectively.

Supplementary Figure 5. Particle size distribution analysis for Li deposited on a) Bare Cu, b) SnO₂, c) ZnO, and d) Al₂O₃ substrates respectively.

Page 20, line 341: Voltage profiles, with lithium deposition capacity plotted only up to 0.4 mAh to accentuate the voltage inflection point, show that the average nucleation overpotentials of lithium on bare copper, ZnO, and SnO₂ are 96 mV (±49 mV), 33 mV (±21 mV), and 54 mV (± 9 mV), respectively, whereas the nucleation overpotential on Al₂O₃ is much higher at 870 mV (± 162 mV) (Figure 5b-5c). These first cycle nucleation overpotential trends are corroborated using at least three cell duplicates (Supplementary Figure 13), and subsequent cycles reveal that the overpotential trends are maintained (Supplementary Figure 14).

Supplementary Figure 13. Voltage profiles showing the first cycle of lithium deposition at 1 mA/cm² on a) bare copper, and ~ 8 nm films of b) SnO₂, c) ZnO, and d) Al₂O₃, showing three cells each.

Supplementary Figure 14. Voltage profiles showing the first three cycles of lithium deposition at 1 mA/cm² on a) bare copper, and ~ 8 nm films of b) SnO₂, c) ZnO, and d) Al₂O₃.

Page 14, line 261: In addition, the TEM results indicate that across all substrates, the SEI formed on lithium has thickness between 15 nm and 18.5 nm and is amorphous, as it does not contain ordered domains (**Figure 4c**). These SEI thicknesses are further corroborated by measurements over 10 distinct SEI domains of Li deposited on each substrate, with the average SEI thickness on bare copper, SnO₂, ZnO, and Al₂O₃ being 17.1 nm (± 5.6 nm), 16.3 nm (± 1.9 nm), 17.7 nm (± 5.0 nm), and 17.4 nm (± 2.8 nm) respectively (**Supplementary Figure 9**).

Supplementary Figure 9. SEI thickness averaged over 10 distinct locations on a lithium particle deposited on the corresponding substrates.

6. The authors suggested that the lithium deposits on SnO₂ (Figure 3) are isolated from the current collector. However, from the SEM image, it is apparent that this morphology is interconnected better than the lithium nuclei plated on other substrates. Moreover, lithium morphologies on the bare copper and ZnO are very similar; while SnO₂ has a very different morphology, it seems that it provides the best current collector coverage. Does the comparison of Li morphologies valid in this case? Was the morphology of a thinner Li layer on SnO₂ observed?

Response: We appreciate the reviewer for this astute observation. While Li grown on SnO₂ appears to have better coverage in Figure 3f, by looking at each particle, it is clear that each Li particle grows away from the current collector surface increasing the likelihood that they become isolated during stripping (Figure R1a). The observation in Figure R1a is supported by a more macroscopic SEM (Figure R1b) which shows that the morphology in Figure R1a is representative. These images are also included in the manuscript as Supplementary Figure 3.

Figure R1. Li particles deposited on 7 nm of SnO₂ shown in a) High magnification , and b) Low magnification.

7. Figure 1 illustrates the different lithium electrodeposition mechanisms as a function of the coating. However, it is confusing since the SEI is not shown on either the active or inactive surfaces. Moreover, it should be emphasised that the SEI's composition and stability could vary between various substrates, especially while some of them alloy lithium.

Response: We agree with the reviewer and have modified the caption to reflect that the SEI is excluded from the schematic for simplicity and that the SEI may vary across different substrates. We have modified the text thus:

Figure 1. Electrical property of ALD-modified copper influences lithium morphology a) Illustration of lithium electrodeposition on substrate modified by a conductive ALD film. Here, multiple nucleation sites accompany the formation of lithium deposits with high packing density (high exposed surface area). b) Illustration of lithium electrodeposition on a substrate modified by a resistive ALD film. Here, few nucleation sites accompany the formation of lithium deposits with low packing density (low exposed surface area). **SEI is not shown for simplicity and could vary across substrates.**

8. Irreversible capacity:

a. How do the non-homogeneous Al_2O_3 coating on Li and Cu affect the SEI formation during the first and further cycles?

Response: We would like to note that the Al_2O_3 film is homogenous, but it has a few pinholes like most thin films. Nonetheless, our early-stage voltage profiles suggest that SEI formation prior to Li nucleation on the Al_2O_3 substrate is minimized in comparison to the other substrates. However, given that the capacity lost to SEI formation prior to Li nucleation is small, the SEI formed across all substrates remains similar as shown in our TEM images in *Figure 4c*.

b. Could a larger exposed surface on the bare copper encourage larger irreversible capacity at the first cycle?

Response: Yes, this could be true if Li particles formed on the bare copper have higher surface area.

9. Page 7, row 125: could the capacity loss be attributed to dead lithium formation and galvanic corrosion in addition to the SEI formation? If SEI formation is suggested as a sole degradation method, further justification is needed.

Response: While galvanic corrosion could play a role in Li loss, previous studies indicate that the timescale under which galvanic corrosion becomes a significant path for lithium loss is about 12 hours,^{17,18} and since our cells are not rested between discharge and charge, it is likely that most of the lithium consumption could be attributed to SEI formation. We agree that dead lithium formation is a potential pathway for capacity loss, so we have modified our text to reflect that:

Revised text on page 7, line 124: The cell with bare copper shows capacity fade in multiple cycle index regions indicating that it experiences cycling instabilities associated with the loss and retrieval of lithium due to SEI formation and possibly dead lithium formation

10. LiNO₃ is used in all the studied systems. Does it react differently with the various surfaces explored in this manuscript?

We appreciate this question posed by the reviewer. To show that LiNO₃ does not decompose differently across the different surfaces explored in our manuscript, we carried out cyclic voltammetry experiments from OCV to 15 mV vs Li⁺/Li. Our results, shown in **Figure R2**, also added to the text as **Supplementary Figure 10**, indicate that LiNO₃ does not decompose differently or react differently across the different surfaces:

Figure R2. Cyclic voltammetry scans between OCV and 15 mV at 1 mV s^{-1} carried out for all substrates using DDN electrolyte. Points 1,2,3, and 4 correspond to cathodic and anodic currents associated with the conversion and alloy reactions of SnO_2 .^{19,20} Point 5 corresponds to Li and ZnO alloy reactions.²¹ Point 6 corresponds to conversion reactions between Li and CuO .²²

11. Figure 2: the range of the Y axis should be minimised to 90-100%, and the size of the symbols should be decreased to emphasise the subtle difference between the efficiency of the cells based on the various coatings.

Response: We have minimized the y-axis to 60-100 % to show the differences in efficiencies of the cells as shown in this new version of Figure 2:

Figure 2. Electrochemical performance of ALD-modified cells demonstrated using Li/Cu cells. All cells were cycled in DDN electrolyte. **a)** CE of Li/Cu cells cycled at a current density of 1 mA/cm² using bare copper, and copper modified with 7-8 nm of SnO₂, Al₂O₃, and ZnO. **b)** CE of Li/Cu cells cycled at a current density of 2 mA/cm² using bare copper, and copper modified with 8 nm of Al₂O₃.

12. The XPS analysis is focused on the C-O and C-F bonds. However, a more detailed analysis of the XPS data is required; specifically, the peaks should be assigned either to the product of the salt or solvents' reduction products.

Response: We thank the reviewer for pointing out the need for clearly identifying the origin of the chemical species found in the SEI. We have specified the potential origin of the C-F bond in the text, but due to the presence of carbon and oxygen in both the salt (LiTFSI) and solvents (DOL/DME), it is difficult to specify the origin of the C-O and Li₂O in the SEI as they could have been formed from the decomposition of either solvent or salt. To clarify, we have modified the manuscript regarding the origin of the C-F bond:

Revised text on page 13, line 235: The high-resolution F 1s peak after 1 minute of sputtering (approximately 2 nm in depth calibrated for SiO₂) indicates the presence of the same C-F bond across all examined samples (Figure 4b). This C-F bonded species is likely a product of salt decomposition in the electrolyte.

a. What was approximately the thickness of the sputtered layer in the XPS experiment?

Response: The approximate thickness calibrated for SiO₂ was 2 nm, and it has now been specified in the text as follows:

Revised text on page 13, line 235: The high-resolution F 1s peak after 1 minute of sputtering (approximately 2 nm in depth calibrated for SiO₂) indicates the presence of the same C-F bond across all examined samples (Figure 4b).

b. Was a reference of the dried electrolyte or electrolyte pristine salt measured?

Response: No, the pristine salt was not measured. Nonetheless, our XPS analysis of the SEI is consistent with numerous studies in which DDN electrolyte was used in lithium batteries^{23,24}, as we had previously stated in our manuscript.

c. Additional XPS measurements are required since further sputtering must be done until the coating or copper signal appears. These additional experiments are essential to the comparison of the SEI composition between the different nucleation layers. Specifically, an XPS measurement and depth profile of the SEI formed on 8 nm Al₂O₃ coating is critical.

Response: We thank the reviewer for suggesting a depth profile experiment, but we would like to point out that a depth profile has limited value for analyzing the SEI because the SEI is heterogenous and made up of distinct domains of different chemical species with different argon gun tolerance. As a result, information collected from depth profile tends to be misleading and often lacks significance. In addition, our TEM images show that the SEI structures have similar thicknesses of around 16 nm, and because the electron escape depth of XPS is ~8 nm, we believe that our XPS analysis of the topmost layer of the SEI reveals sufficient chemical details about the full depth of the SEI structure. Finally, we have performed sputtering experiments to reveal the Al₂O₃ and Cu substrates as the reviewer suggested.

Revised text on page 13, line 224: The absence of Al, Sn, and Zn signals prior to and after three minutes of sputtering indicates that lithium is deposited atop the Al₂O₃, SnO, and ZnO nucleation films, respectively (Figure 4a). By sputtering 0.1 mAh/cm² of Li deposited on Al₂O₃, we observe a reduction in SEI-specific elements (C, N, and F) and a gradual increase in Al and Cu signals (Supplementary Figure 7), indicating that Li deposits atop our ALD films.

Supplementary Figure 7. Sputter profile of 0.1 mAh/cm² of Li, sputtered at 2 nm/min calibrated for SiO₂. a) Atomic concentration as a function of sputter time. b) Atomic ratio as a function of sputter time.

13. In Figures 3i-3k, the lighter contrasted features were assigned to "dead lithium". Could these features also be identified as a very thick SEI or non-reacted electrolyte layer?

Response: Yes, these features could also be identified as SEI formed from continuous electrolyte decomposition. To reflect this, we have modified the text thus:

Revised text on page 12, line 201: The lighter structures can be identified as **accumulated SEI** or dead, electrochemically-irretrievable lithium as has been previously demonstrated.

14. Supplementary Figure 3b, a discussion of the S 2p spectra is missing.

Response: We have added the following discussion:

Revised text on page 13, line 240: **All substrates reveal O 1s spectra containing C-O and Li₂O, and weak S 2p spectra (Supplementary Figures 8a-8b).**

15. Was the amount of "dead Li" measured or estimated?

Response: No, the quantity of dead Li was not directly measured, but the CE data (lower CE in the presence of similar SEI thicknesses corresponds to more dead Li), the electrochemical impedance spectroscopy data (higher interfacial impedance is indicative of a more resistive interface, as is the case with more dead Li), and the SEM images after 50 cycles all support the hypothesis that less dead Li is formed atop Al₂O₃-modified substrates. These supportive data are all described in the manuscript.

16. Impedance and electronic conductivity measurements:

a. Please provide further experimental details on the impedance measurements.

b. How was the charge transfer impedance estimated? Please provide the equivalent circuit used for the fitting and the fitting parameters.

c. How many EIS experiments were conducted? What is the error for the charge transfer impedance? Are the differences in charge transfer impedance and electrical conductivity statistically significant?

Response: To address the comments raised in parts a, b, and c, we have added the following new figure section (equivalent circuit), table (fitting parameters), and additional data points to the manuscript.

Revised text:

Supplementary Figure 6. Electrochemical impedance of Li deposited after 50 electrochemical cycles using bare copper and copper modified with ~8 nm of ALD films. Solid black lines represent EIS fits obtained using the equivalent circuit at the bottom of the figure. Parameters obtained from the fits are presented in supplementary table 1.

Supplementary Table 1. Electrochemical impedance of Li deposited after 50 electrochemical cycles using bare copper and copper modified with ~8 nm of ALD films. R_{int} was obtained from the sum of R2 and R3, with both quantities obtained by fitting the equivalent circuit shown in Supplementary Figure 2. Each fit is carried out using two unique cell replicates.

Substrate	R1 (Ohms)	C2 (F)	R2	C3	R3	Q1 ($F \cdot s^{(a-1)}$)	a1 No unit	R_{int}
Bare Cu	23.50	24.07E-06	4.49	7.30E-06	1.99	0.065	0.47	6.48
	6.97	4.54E-06	2.52	12.84E-06	7.10	0.048	0.43	9.62
SnO ₂ (7nm)	12.50	2.74E-06	2.32	18.96E-06	5.12	0.077	0.46	7.44
	7.71	12.14E-06	6.42	3.473E-06	2.52	0.047	0.46	8.94
ZnO (7 nm)	10.60	7.45E-06	1.99	27.22E-06	4.16	0.089	0.52	6.15
	10.40	20.23E-06	4.55	6.17E-06	2.30	0.053	0.46	6.85
Al ₂ O ₃ (8 nm)	9.75	24.57E-06	2.49	1.781	0.70	0.106	0.56	3.19
	7.07	26.57E-06	3.32	7.311E-06	1.57	0.067	0.57	4.89

d. What is the physical meaning of the solution resistance?

Response: The solution resistance represents the resistance to ion transport within the liquid electrolyte.²⁵ It could indicate differences in intrinsic ionic conductivity between electrolytes or changes in the ionic conductivity of a particular electrolyte during battery cycling. We have also described the physical significance of solution resistance in our manuscript.

Revised text on page 12, line 210: Solution resistance represents resistance to ion transport within the liquid electrolyte,²⁵ so the low solution resistance observed with Al₂O₃ substrates indicates a reduction in electrolyte consumption, while the low SEI charge transfer resistance is evidence of the lower prevalence of dead lithium.

e. The axes of the Nyquist plots should be of the same size; otherwise, the shape of the spectra is distorted and loses physical meaning.

We thank the reviewer for this note and have corrected our EIS plots to reflect this as shown below:

Supplementary Figure 6. Electrochemical impedance of Li deposited after 50 electrochemical cycles using bare copper and copper modified with ~8 nm of ALD films. Solid black lines represent EIS fits obtained using the equivalent circuit at the bottom of the figure. Parameters obtained from the fits are presented in supplementary table 1.

f. Please discuss the effect of the surface area of the deposited Li on the measured impedance. Were the ratios between the charge transfer impedance values validated using another method?

Response: The ratios between the charge transfer impedance values were not measured using another method, but we think that the consistency in charge transfer measurements, CE measurements, and SEM images indicates the reliability of our measurements. We agree that the surface area of Li is inversely proportional to the measured impedance; however, because the measured impedance also depends on dynamic quantities like SEI thickness and electrolyte conductivity, it is difficult to infer Li surface area directly from impedance measurements. We note the dependence of impedance on surface in the main text as follows:

Revised text on page 12, line 211: The low solution resistance observed with Al₂O₃ substrates indicates a reduction in electrolyte consumption, while the low SEI charge transfer resistance is evidence of the lower prevalence of dead lithium. While an inverse relationship between charge transfer resistance and

lithium surface area exists, we assume that the lithium particles across all substrates have similar surface areas after the 50th cycle due to their similar SEM morphologies, making our estimates of charge transfer resistance reasonable for identifying the prevalence of dead Li.

17. Page 12 row 216: could the Al signal be a result of Al corrosion?

Response: It is very unlikely that the observation is caused by Al corrosion because Al corrosion in the presence of Li would lead to a lithium-aluminate product that would have a lower binding energy than what we observe in *Figure 4A*. Instead, we think that the partial coverage of Al₂O₃ by Li after Li electrodeposition results in the detection of Al during XPS analysis.

18. Overpotential

a. The authors attributed the larger nucleation overpotential to higher electronic resistance of the substrate, in contrast with the common understanding in the literature, attributing larger overpotential to more heterogeneous metal plating. Please contrast these findings with several studies in the literature.

Response: The relationship we posed between nucleation overpotential and substrate resistance is consistent with the literature. Here, we are saying that an increase in substrate resistance reduces the number of nucleation spots on the substrate, thereby increasing the local current density of lithium ions and increasing nucleation overpotential. In the literature, an analog of this relationship has been presented in several studies in which they show that by fixing the substrate (copper) and increasing current density, nucleation overpotential increases.^{26,27} The only difference between our work and the literature is that our resistive substrates promote low surface area lithium particles even at high overpotentials because of the diffusion model that we present in the paper.

b. Was a similar overpotential ratio observed in the following cycling of the half and full cells?

Response: Yes. Evidence of this has been added to the manuscript as shown here:

Revised text on page 20, line 341:

Voltage profiles, with lithium deposition capacity plotted only up to 0.4 mAh to accentuate the voltage inflection point, show that the average nucleation overpotentials of lithium on bare copper, ZnO, and SnO₂ are 96 mV (±49 mV), 33 mV (±21 mV), and 54 mV (± 9 mV), respectively, whereas the nucleation overpotential on Al₂O₃ is much higher at 870 mV (± 162 mV) (**Figure 5b-5c**). These first cycle nucleation overpotential trends are corroborated using at least three cell duplicates (**Supplementary Figure 13**), and subsequent cycles reveal that the overpotential trends are maintained (**Supplementary Figure 14**).

Supplementary Figure 13. Voltage profiles showing the first three cycles of lithium deposition at 1 mA/cm² on a) bare copper, and ~ 8 nm films of b) SnO₂, c) ZnO, and d) Al₂O₃.

19. The authors demonstrated a mushroom-like Li growth on the Al₂O₃ coated substrates. Were any morphological or composition differences between the surface in the centre of the round Li deposits and on the edges found?

Response: The reviewer poses a very important point here. Yes, morphological differences exist between the center of the deposits (where nucleation occurs on bare copper) and at the edge of the deposits, where growth occurs on Al₂O₃. We have added this note to the manuscript:

Revised text on page 25, line 425: By depositing lithium atop the patterned substrate in **Figure 5i**, we observe lithium morphologies that originate from the active surfaces and grow radially outward into flat, planar, pancake-like deposits (**Figure 5j and Supplementary Figure 21**). By observing the morphologies closely in **Supplementary Figure 21**, it is evident that the particles formed on bare Cu are significantly smaller than the lithium deposits that grow atop Al₂O₃, supporting the diffusion model.

20. A coating of hafnium-ethylene glycol (HfEG) was briefly introduced. Was this type of coating used before in alkali metal batteries? How does this coating compare with the 8nm Al₂O₃ coating?

Response: This HfEG coating has not been used in Alkali metal batteries prior to this report. We only present the coating to show that the relationship between resistance and lithium morphology is not restricted to Al₂O₃, as the HfEG coating is expected to possess similar electrical characteristics as Al₂O₃.

21. The performance of the 8 nm Al₂O₃ coating 8nm was utilised in Cu//NMC 532 pouch cells (Figure 6k and 6l). The scalability of the coating method is impressive; however, the slopes of the coated and bare copper-based cells are very similar, and it seems that the difference is caused merely by the capacity drop at the first cycle. Please discuss the various possible degradation mechanisms and the contribution of the Al₂O₃ coating in the experiment presented in Figure 6k.

Response: We acknowledge the reviewer's observations. We want to point out that while the slopes of the discharge capacity plots may appear similar, they are actually quite different, with the Al₂O₃ cell showing a lower slope. We support this conclusion by pointing out that the difference in discharge

capacity between both cells is about 12.2 % in the 2nd cycle while the difference in discharge capacity is 21.6 % in the 100th cycle (**Figure R3**). In addition, our film reduces the number of cycles required to form a stable SEI, as observed from the 2nd cycle CE improvement from 84.08 % in the control cell to 95.87 % in the Al₂O₃ cell.

Figure R3. Normalized discharge capacity of anode-free Cu||NMC 532 pouch cells showing the differences in capacity between the bare copper substrate and Al₂O₃-modified substrate after the 2nd and 100th discharge cycles.

We have modified our text to make the performance distinctions between Al₂O₃ and bare copper clearer.

Revised text on page 27, line 473: The Al₂O₃-modified cell outperforms the bare copper cell because it attains a Coulombic efficiency of 95.87 % in the 2nd cycle while the cell with bare copper only attains a Coulombic efficiency of 84.08 % in the 2nd cycle (**Figure 6I**). In addition, the Al₂O₃-modified cell maintains a high discharge capacity over extended cycles with a 12.2 % and 21.6 % improvement in capacity over the bare copper cell in the 2nd and 100th cycles respectively, showing the high cycling reversibility of Al₂O₃-modified cells.

Reviewer #2 (Remarks to the Author):

The authors present a compressive study of the effect of various passivation strategies on the Li deposit morphology and electrochemical performance. The authors show that these effects are also present when the electrolyte is varied. The work presented by Oyakhire and co-workers is well structured and clearly presented, while the results provide enough novelty and thoughtful discussion, therefore, the manuscript, is recomended for publication in Nature Communications. Below are some questions that can be considered by the authors to further be addressed in their manuscript:

We appreciate the reviewer's belief in the comprehensiveness of our work and have addressed their suggestions. We believe that the reviewer's suggestions make our work better.

1. The voltage profiles for initial cycles should be reported (maybe in SI), as this would clarify some of the questions raised and also the discussions made by the authors.

Response: We appreciate this point from the reviewer and have added the following voltage profiles for the initial cycles to the SI:

Revised text on page 20, line 341:

Voltage profiles, with lithium deposition capacity plotted only up to 0.4 mAh to accentuate the voltage inflection point, show that the average nucleation overpotentials of lithium on bare copper, ZnO, and SnO₂ are 96 mV (± 49 mV), 33 mV (± 21 mV), and 54 mV (± 9 mV), respectively, whereas the nucleation overpotential on Al₂O₃ is much higher at 870 mV (± 162 mV) (Figure 5b-5c). These first cycle nucleation overpotential trends are corroborated using at least three cell duplicates (Supplementary Figure 13), and subsequent cycles reveal that the overpotential trends are maintained (Supplementary Figure 14).

Supplementary Figure 14. Voltage profiles showing the first three cycles of lithium deposition at 1 mA/cm² on a) bare copper, and ~ 8 nm films of b) SnO₂, c) ZnO, and d) Al₂O₃.

2. In Figure 3, what is preventing Li deposits to grow in length towards the CE on the copper modified with Al₂O₃? Please discuss.

Response: Li grows in length towards the counter electrode (CE), but its upward growth rate is slower than its lateral growth rate. The preferential lateral growth across the current collector is likely due to faster diffusion of ions across the inactive sites on the current collector, due to a build-up of diffusion fields perpendicular to the current collector surface. We provide a full discussion of this effect in our manuscript:

“Our results suggest that, atop resistive substrates, the likelihood for electron transport from the external circuit is reduced and possibly restricted to pinholes and defects, thereby limiting the nucleation of lithium to the few defect sites on the substrate. We propose that lithium deposition atop resistive

substrates proceeds according to the model illustrated in Figure 5h. The defect sites and pinholes in the resistive substrates represent the active sites while the other parts of the substrate, in which electron transport is prohibitive, are classified as inactive sites (Figure 5h). After the desolvation of lithium ions from solvent molecules, lithium ions will migrate towards the surface of the current collector; however, nucleation will only occur at the active surface sites. Subsequently, lithium ions that impinge on the inactive surfaces of the current collector diffuse towards the active sites, to access electrons via lithium metal that is nucleated at the active surface sites (Figure 5h).

This diffusion driven growth of lithium at steady state is fundamentally similar to the analytical treatment of diffusion-controlled currents at electrodes surfaces that contain electrically active and inactive areas.^{42,43} Under short-time scales, it is derived analytically that each electrically active spot generates a linear diffusion field of species from the solution phase (perpendicular approach of species towards the electrode surface).^{42,43} At longer time scales, the active surfaces become occluded by a diffusion layer, and as a result, diffusion becomes dominated by radial transport (non-perpendicular approach) of species via the inactive surface, towards the active electrode surface.^{42,43} In a system like ours, where the electrically active surfaces are hypothesized to be small film defects, those surfaces behave like ultramicroelectrodes (UMEs), in which radial diffusion of species towards the active surfaces dominates linear diffusion even at very short time scales. A mathematical justification for this behavior is presented in Supplementary Note 2. This preference for lateral growth over vertical growth on resistive substrates is demonstrated using finite element simulations in Supplementary Figure 19. We propose that the nucleation of lithium at the few active sites and the radial growth of lithium via diffusion of lithium ions from the inactive surfaces on the current collector are responsible for the sparse and planar morphology of lithium observed atop resistive substrates.”

3. What is the initial resistance of the investigated cells? Regarding the Figure S2, the higher amount of accumulated dead Li in the cell, would not result in a higher resistance of such cells? Table 1 can be further supplemented with such information.

Response: We thank the reviewer for this question and have supplemented our manuscript with information on the more information regarding the resistive and capacitive components of our EIS plots after by fitting the plots to a well-established equivalent circuit. We would also like to note that due to differences in initial Li surface area across all the substrates, impedance measurements will not suffice for estimating the interfacial charge transfer resistance. As a result, all our impedance measurements were carried out after 50 cycles, when the Li surface area appears similar across all substrates.

Revised text on page 12, line 213:

While an inverse relationship between charge transfer resistance and lithium exists, we assume that the lithium particles across all substrates have similar surface areas after the 50th cycle due to their similar SEM morphologies, making our estimates of charge transfer resistance reasonable for identifying the prevalence of dead Li.

Supplementary Figure 6. Electrochemical impedance of Li deposited after 50 electrochemical cycles using bare copper and copper modified with ~8 nm of ALD films. Solid black lines represent EIS fits obtained using the equivalent circuit at the bottom of the figure. Parameters obtained from the fits are presented in supplementary table 1.

Supplementary Table 1. Electrochemical impedance of Li deposited after 50 electrochemical cycles using bare copper and copper modified with ~8 nm of ALD films. R_{int} was obtained from the sum of R2 and R3, with both quantities obtained by fitting the equivalent circuit shown in Supplementary Figure 2. Each fit is carried out using two unique cell replicates.

Substrate	R1 (Ohms)	C2 (F)	R2	C3	R3	Q1 ($F*s^{(a-1)}$)	a1 No unit	R_{int}
Bare Cu	23.50	24.07E-06	4.49	7.30E-06	1.99	0.065	0.47	6.48
	6.97	4.54E-06	2.52	12.84E-06	7.10	0.048	0.43	9.62
SnO ₂ (7nm)	12.50	2.74E-06	2.32	18.96E-06	5.12	0.077	0.46	7.44
	7.71	12.14E-06	6.42	3.473E-06	2.52	0.047	0.46	8.94
ZnO (7 nm)	10.60	7.45E-06	1.99	27.22E-06	4.16	0.089	0.52	6.15

	10.40	20.23E-06	4.55	6.17E-06	2.30	0.053	0.46	6.85
Al ₂ O ₃ (8 nm)	9.75	24.57E-06	2.49	1.781	0.70	0.106	0.56	3.19
	7.07	26.57E-06	3.32	7.311E-06	1.57	0.067	0.57	4.89

4. The increase of the electrolyte resistance was assigned on the changes in the electrode morphology and the subsequent reaction with the electrolyte components. Can you subtract the effect of the electrode's composition and morphology from the cells cycled for 50 cycles, by measuring the electrolyte resistance with blocking electrodes? This would strengthen the statement at page 11, line 201.

Response: We agree with the reviewer that the solution resistance that we measured is convoluted by other factors. We also think that measuring the electrolyte resistance after 50 cycles will be somewhat of a challenge because after 50 cycles, most of the electrolyte left in the cell will be difficult to recover. As a result, we have restructured our initial statement to capture the uncertainties around the relationship between our measured impedance and the electrolyte resistance:

Revised text on page 12, line 211: The low solution resistance observed with Al₂O₃ substrates indicates a reduction in electrolyte consumption, while the low SEI charge transfer resistance is evidence of the lower prevalence of dead lithium. While there is an inverse relationship between charge transfer resistance and lithium surface area, we assume that the lithium particles across all substrates have similar surface areas after the 50th cycle due to their similar SEM morphologies, making our estimates of charge transfer resistance reasonable for identifying the prevalence of dead Li.

5. Regarding the studies on deposited Li, did you investigate the morphology of the Li nucleates and their location, when a smaller amount of Li is deposited as compared to the 1mAh/cm²? Please discuss.

We thank the reviewer for this question. We already investigated the morphologies of 0.05 mAh/cm² Li nucleates and their location as shown in supplementary Figure 14, and we found that selective nucleation and growth occurs in the resistive substrates, while non-preferential nucleation occurs in the conductive substrates, as shown and discussed in the main text:

Revised text:

This result suggests that the sparse particle density of lithium on Al₂O₃, as observed in **Figure 3d**, is caused by a low number of available sites for lithium nucleation. Using a smaller lithium capacity (0.05 mAh/cm²) closer to the nucleation regime, we also observe much larger lithium deposits (approximately 10 times in diameter) atop Al₂O₃ substrates compared to SnO₂ and ZnO substrates (**Supplementary Figure 15**). The larger lithium deposits observed on Al₂O₃ after the onset of nucleation further support the argument that the reduction of electrical conductivity in Al₂O₃ reduces the number of nucleation sites, limiting the sites of Li growth to a smaller number of existing lithium nuclei.

Supplementary Figure 15. Top-view SEM images of 0.05 mAh/cm^2 of lithium plated at 0.02 mA/cm^2 on all substrates.

6. Regarding the XPS investigations, these investigations were carried out for the samples after 1 mAh/cm^2 was deposited. How is the outcome of the XPS measurements in accordance with the statement of the EIS measurements?

Response: We thank the reviewer for asking this question. Our EIS measurements were carried out after 50 cycles to examine the extent of electrolyte consumption and SEI buildup in our cell, while the XPS analysis was carried out after 1 cycle of electrodeposition to probe the decomposition products formed atop freshly deposited Li. We refrain from using XPS to examine the surface of Li after 50 cycles, because as our SEM images show, the surface of Li becomes very heterogenous, making it difficult to extract meaningful XPS insights. We also do not carry out EIS on Li deposited after one cycle because there are huge variations in the Li particle surface area formed across substrates, making EIS measurements difficult to rationalize. As a result, we augment our XPS with TEM results and augment our EIS results with SEM images as shown in our manuscript.

7. What is the effect of the chemical composition of the SEI on low amount of Li deposited (maybe 0.1 mAh/cm^2) during an initial cycle? Then these XPS results would be complementary to the cryo-TEM investigations.

Response: Based on past studies, we expect the SEI formed on 0.1 mAh of Li to be similar to the SEI formed on 1 mAh of Li. This similarity between SEIs is expected because the SEI is formed from spontaneous reactions between Li and the electrolyte (at capacities $<0.1 \text{ mAh}$) and significant changes in

SEI chemistry occur would only occur at large timescales (>2h).²⁸ We have added a statement to reflect the complementary nature of cryo-TEM and XPS:

Revised text on page 14, line 247: While the SEI chemical composition is critical for cell lifetime and performance, its structure and thickness also play a key role in the ease of ion mobility across electrode-electrolyte interfaces. Using fully developed cryogenic-TEM (cryo-TEM) methods,^{13,29} we preserve the native SEI and examine its structure and thickness on 0.1 mAh/cm² of plated lithium. A relatively small capacity of lithium is used to ensure electron transparency during cryo-TEM analysis. SEI information collected using cryo-TEM provides complements the chemical information collected from XPS analysis even though Li deposition capacities are different.³⁰

8. With respect to the statement at line 246, are there other effects that degrade the cell performance with time for various substrates? What about the interface Cu|electrolyte? (where the Cu substrate contains Li deposits, thus this substrate would be at the potential of metallic Li or a mixed potential)

Response: The reviewer raises another excellent point. We agree that the Cu interface plays a significant role in long term galvanic corrosion in the battery. Corrosion events are the subject of further investigation in our lab. Nevertheless, we have modified our text to reflect the possibility of other factors that could affect the long-term performance of the cells:

Revised text on page 11, line 181: It is noteworthy that the microstructure of lithium on Al₂O₃ appears more compact and interlocked than does lithium on the other substrates, possibly indicating better contact with the copper foil. Lithium deposits that are in good contact with copper are reportedly more electrochemically retrievable,^{31,32} consistent with the improved performance observed in the presence of Al₂O₃. While there is clear evidence that the different Li morphologies formed on these substrates contribute to differences in performance, it is possible that factors such as galvanic corrosion also influence performance differences on a long-term scale

9. Regarding Figure 4d-f, in the case of lithiophilic coatings (i.e. SnO₂ and ZnO) the formation of Zn and Li₂O would occur prior the Li deposition. When further Li is deposited, LiZn will form and this will be also reversible. Have you carried out cyclic voltammetry to 15mV in order to further investigate if these reactions are not occurring? This should be carefully investigated, as also in Figure 5b there is an deflection (prior the inflection point and well above 0V) observed for Cu, ZnO@Cu and SnO₂@Cu as compared to Al₂O₃@Cu.

Response: We appreciate this suggestion by the reviewer and have carefully carried out new CV experiments between open circuit voltage (OCV) and 15 mV for each substrate. For SnO₂, we observe cathodic and anodic currents associated with electrolyte decomposition, and in addition, reversible conversion and alloying peaks (depicted as 1,2,3, and 4 in Supplementary Figure 10), with the reversibility of the reactions explaining why we observe Sn in its native oxide state using XPS. For bare Cu, we observe similar electrolyte reduction peaks and a peak commonly associated with the formation of Li₂O from CuO (depicted as peak 6 in Supplementary Figure 10). For Al₂O₃, we observe only the background currents associated with the decomposition of the electrolyte, and notably, the currents are smallest for the Al₂O₃ substrate, in agreement with its high electrical resistance. For ZnO, in addition to electrolyte decomposition, we observe an irreversible alloying reaction (depicted as 5 in Supplementary Figure 10), suggesting that the ZnO substrate forms LiZn prior to electrodeposition. However, our XPS

binding energy results show that Zn maintains its oxidation state, suggesting that majority of the ZnO film remains unreacted. We have added the CV results into our manuscript with the addition of new Supplementary Figure 10, as well as explanatory text (see response to comment #10 below):

Supplementary Figure 10. Cyclic voltammetry scans between OCV and 15 mV at 1 mV s^{-1} carried out for all substrates using DDN electrolyte. Points 1,2,3, and 4 correspond to cathodic and anodic currents associated with the conversion and alloy reactions of SnO_2 .^{19,20} Point 5 corresponds to Li and ZnO alloy reactions.²¹ Point 6 corresponds to conversion reactions between Li and CuO .²²

10. The authors state that such a behavior, as the one discussed at line 278-280, is due to the difference in kinetic affinity for Li towards electrolyte instead of lithiphilic layer. It is not clear what Li is expected at 15mV on the surface of Cu substrate. Please discuss.

Response: We appreciate the comment from the reviewer. What we meant was that electrolyte decomposition reactions may have been more favorable than reactions between Li ions and the corresponding substrates. To clarify this, we have restructured that part of our text:

Revised text on page 18, line 289: To understand the lithium-current collector interface, we examine the chemical nature of the substrates just before the onset of lithium nucleation. In our previous study, we reported that TiO_2 reacts with lithium to form a Li_xTiO_2 alloy prior to the onset of electrodeposition.³³ Changes to the current collector prior to nucleation could elicit differences in lithium morphology

especially if a lithium alloy is formed on the current collector. To investigate the interface between lithium and the ALD-modified current collectors, we hold the Li/Cu cells at 15 mV, just above the nucleation potential for lithium, for 8 hours. Following this voltage hold, we carry out XPS on the ALD-modified copper substrates. From **Figure 4d**, it is evident that there is no reaction between Al_2O_3 and lithium ions prior to electrodeposition because the binding energy of Al 2p (74.6 eV) remains consistent with Al in the bonding environment of Al_2O_3 .³⁴ This result is not surprising because the lithium and Al_2O_3 reaction is known to have a high energy barrier at room temperature.³⁵ **Figure 4d** also reveals that SnO_2 and ZnO do not react with Li prior to nucleation, as indicated by the binding energy of Sn 3d 5/2 and Zn 2p 3/2, which show up at 486.6 eV and 1022.6 eV, respectively, suggesting the presence of SnO_2 ³⁶ and ZnO.³⁷ This finding is surprising because the reaction between lithium and SnO_2 or ZnO should be energetically and electrochemically favorable.³⁸ Using cyclic voltammetry (CV), we find that conversion and alloy reactions occur on the SnO_2 and ZnO substrates (**Supplementary Figure 10**). However, because the XPS binding energy of Sn and Zn indicate the presence of their corresponding oxides, it is likely that a large fraction of the ALD films do not react with Li ions during the potentiostatic hold at 15 mV. Because there is no strong evidence for direct reaction between Li ions and any of the three metal oxide films, the observed differences in lithium morphology and performance of cells appear to be the result of differences in the intrinsic properties of the ALD films used to modify the copper current collector.

11. What is the electrical resistivity of the 8nm ALD coatings on Copper that were investigated in this manuscript? Are then the results in accordance to those measured for 50nm thick coatings?

Response: The resistivity of the Al_2O_3 film remains constant while its resistance increases with increasing thickness. We used the 50 nm film for resistivity measurements to ensure reproducibility across film domains in our four-point probe experiments. We have modified our text to reflect this.

Revised text on page 19, line 330: We measure the resistivity of 50 nm of each metal oxide film used in this study, grown on Si wafers. We used 50 nm films to ensure reproducibility across film domains in our experiments.

12. Regarding the existence of pin holes, have you tried to deposit Al_2O_3 at higher temperature than 120°C?

Response: We did not deposit alumina at temperatures higher than 120°C, but we expect that 8 nm thin alumina films will possess pinholes regardless of deposition temperature because it takes significantly higher thicknesses for films to become pinhole-free.^{39,40} This relationship between pinhole density and film thickness was previously cited in our manuscript, as shown below:

Text on page 24, line 420: To verify this hypothesis, we deposit 50 nm of Al_2O_3 atop a copper substrate. Such a high thickness of Al_2O_3 is expected to reduce the likelihood of forming pinholes in the film.⁴¹

13. Would the effect of a surface modified 8nm Al_2O_3 Cu substrate be similar to that observed for the 50nm Al_2O_3 substrate, when Li will be deposited?

Response: While the effects of the film on Li growth structure are the same as shown in our SEM figures, the overpotential required to deposit Li on 50 nm Al_2O_3 is prohibitively large and outside the safe range

of our battery cyclers. This is particularly true because the pinhole density of a 50 nm Al_2O_3 film is very small,⁴¹ making it difficult to nucleate and grow Li particles.

14. There are some early reports that state that Al_2O_3 prepared at low temperature and in a thin layer is also partially conductive for Li-ions. Is then, the formation of the Li deposits mainly related to the existence of defects in the coating?

Response: Yes, we think that this is correct, and that Li nucleates within defects in our film This mode of nucleation is reflected within our text thus:

Revised text on page 24, line 416: We propose that the nucleation of lithium at the few active sites (defects) and the radial growth of lithium via diffusion of lithium ions from the inactive surfaces on the current collector are responsible for the sparse and planar morphology of lithium observed atop resistive substrates.

15. Regarding the use of Li||Cu cells and Cu||NMC cells, could you compare the capacity retention that is recorded with these cells? From the comparison of only the Coulombic efficiency, one cannot get the whole picture of the negative electrode's performance. Even with the negative electrode's substrate modification, why is there a capacity loss observed in full cells?

Response: We believe that Figure 6k reveals the capacity trends in our full cell and from those trends, it is evident that the Al_2O_3 modified cell improves the capacity that is retained during cycling. We support this by pointing out that the difference in discharge capacity between both cells is about 12.2 % in the 2nd cycle while the difference in discharge capacity is 21.6 % in the 100th cycle (Figure R4). In addition, our film reduces the number of cycles required to form a stable SEI, as observed from the 2nd cycle CE improvement from 84.08 % in the control cell to 95.87 % in the Al_2O_3 cell.

Figure R4. Normalized discharge capacity of anode-free Cu||NMC 532 pouch cells showing the differences in capacity between the bare copper substrate and Al_2O_3 -modified substrate after the 2nd and 100th discharge cycles.

Finally, regarding the reviewer's comment about capacity loss in our cells, we would like to note that because the CE of our modified substrates are not exactly 100 % (~ 99.5 %), there will be compounded loss of active lithium during each electrodeposition and electro-dissolution cycle at the anode, resulting in the capacity loss observed in **Figure R4**.

We have modified our text to make the performance distinctions between Al₂O₃ and bare copper clearer.

Revised text on page 27, line 473: The Al₂O₃-modified cell outperforms the bare copper cell because it attains a Coulombic efficiency of 95.87 % in the 2nd cycle while the cell with bare copper only attains a Coulombic efficiency of 84.08 % in the 2nd cycle (**Figure 6I**). In addition, the Al₂O₃-modified cell maintains a high discharge capacity over extended cycles with a 12.2 % and 21.6 % improvement in capacity over the bare copper cell in the 2nd and 100th cycles respectively, showing the high cycling reversibility of Al₂O₃-modified cells.

Reviewer #3 (Remarks to the Author):

In this manuscript, the author deeply studied the changes of lithium deposition morphology after modifying copper collector by ALD deposition film, and explored the difference between conductive film and resistive film in the process of lithium deposition by using a variety of characterization methods. The Al₂O₃-modified cells show significantly higher cycling stability, reaching a cycle index of 390 at an average CE of 96 % before showing signs of capacity fade. Therefore, this manuscript is acceptable after addressing the following issues.

We really appreciate the reviewer's vote of confidence in our work, and we think that their suggestions have made our work better.

1) In Figure 5b-5c, the nucleation overpotential of lithium on ZnO and SnO₂ is larger than that on bare copper. ZnO and SnO₂ are conductive nanofilms, should have higher affinity for lithium deposition. Please explain why is the nucleation overpotential larger?

Response: We appreciate the reviewer for this point. Even though ZnO and SnO₂ are conductive nanofilms that have a high affinity for lithium deposition, they present barriers for electron transfer at the electrode-electrolyte interface. This barrier to electron transfer is subdued in copper, given that it is a metal. And because nucleation overpotential is a combination of a substrate's affinity for lithium and barrier to electron transfer, it is very likely that the ZnO and SnO₂ films would show a higher nucleation overpotential than Cu when they possess electron transfer barriers. However, upon carrying out multiple experiments, we found that the SnO₂ and ZnO substrates have lower nucleation overpotential on average than bare Cu. Modifications to our text and figures are shown here:

Revised text on page 20, line 340: To further explore this result, we deposit 1 mAh/cm² of lithium at 1 mA/cm² on all substrates: bare copper, and copper modified with 8 nm of Al₂O₃, 7 nm of ZnO, and 7 nm of SnO₂. Voltage profiles, with lithium deposition capacity plotted only up to 0.4 mAh to accentuate the voltage inflection point, show that the average nucleation overpotentials of lithium on bare copper, ZnO, and SnO₂ are 96 mV (±49 mV), 33 mV (±21 mV), and 54 mV (± 9 mV), respectively, whereas the nucleation overpotential on Al₂O₃ is much higher at 870 mV (± 162 mV) (Figure 5b-5c). These first cycle nucleation overpotential trends are corroborated using at least three cell duplicates (Supplementary Figure 13), and subsequent cycles reveal that the overpotential trends are maintained (Supplementary Figure 14).

Figure 5. Understanding the intrinsic differences between bare copper and copper modified with ~8 nm of ALD films. a) Normalized electrical resistivity of ALD-modified substrates. b) Voltage profiles showing the first cycle of lithium deposition at 1 mA/cm² on bare copper and ALD-modified copper. c) Extracted nucleation overpotential of lithium plated at 1 mA/cm² on bare and ALD-modified copper substrates averaged over three cells. d-g) Top-view SEM images of 1 mAh/cm² of lithium plated at 1 mA/cm² on 1 nm, 2 nm, 4 nm, and 8 nm of Al₂O₃-modified copper, respectively. h) Illustration of lithium metal deposition on a resistive substrate. i) Optical image of a 50 nm Al₂O₃-modified copper substrate with 25 μm sized holes that expose the underlying copper substrate. j) SEM image (approximately 30-

degree tilt) of 0.5 mAh/cm^2 of lithium deposited at 1 mA/cm^2 atop the patterned substrate shown in panel 5i.

Supplementary Figure 13. Voltage profiles showing the first cycle of lithium deposition at 1 mA/cm^2 on a) bare copper, and $\sim 8 \text{ nm}$ films of b) SnO₂, c) ZnO, and d) Al₂O₃, showing three cells each.

Supplementary Figure 14. Voltage profiles showing the first three cycles of lithium deposition at 1 mA/cm^2 on a) bare copper, and $\sim 8 \text{ nm}$ films of b) SnO₂, c) ZnO, and d) Al₂O₃.

2) In Figure 6k, the difference in capacity retention between Al₂O₃-modified copper and bare copper is mainly in the first cycle. Does it mean that there is not much difference between bare copper and Al₂O₃-modified copper after the formation of the SEI on the bare copper?

Response: We acknowledge the reviewer's astute observations. We want to point out that while the slopes of the discharge capacity plots look similar, they are quite different, with the Al₂O₃ cell showing a lower slope. We support this by pointing out that the difference in discharge capacity between both cells is about 12.2 % in the 2nd cycle while the difference in discharge capacity is 21.6 % in the 100th cycle (Figure R5). In addition, our film reduces the number of cycles required to form a stable SEI, as observed from the 2nd cycle CE improvement from 84.08 % in the control cell to 95.87 % in the Al₂O₃ cell.

Figure R5. Normalized discharge capacity of anode-free Cu||NMC 532 pouch cells showing the differences in capacity between the bare copper substrate and Al₂O₃-modified substrate after the 2nd and 100th discharge cycles.

We have modified our text to make the performance distinctions between Al₂O₃ and bare copper clearer.

Revised text on page 27, line 473: The Al₂O₃-modified cell outperforms the bare copper cell because it attains a Coulombic efficiency of 95.87 % in the 2nd cycle while the cell with bare copper only attains a Coulombic efficiency of 84.08 % in the 2nd cycle (**Figure 6l**). In addition, the Al₂O₃-modified cell maintains a high discharge capacity over extended cycles with a 12.2 % and 21.6 % improvement in capacity over the bare copper cell in the 2nd and 100th cycles respectively, showing the high cycling reversibility of Al₂O₃-modified cells.

3) Resistive nanofilm may play a role in low current, but can they be better than conductive nanofilms at higher current? It is hoped that the rate performance of Li/Cu cells using bare copper and copper modified with 7-8 nm of SnO₂, Al₂O₃, and ZnO can be compared.

Response: The reviewer raises an excellent point here. While we show that the resistive films perform better than conductive films at low and medium current densities, our experiments suggest that the overpotential required for cycling cells at higher current densities ($> 4\text{mA cm}^{-2}$) are outside of the safe operating limits of our battery cyclers. This suggests that the benefits of the resistive films are momentarily constrained to lower current densities. We have introduced the following modification to our text to reflect this:

Revised text on page 26, line 463: In addition, the electrolyte-agnostic morphology of lithium atop Al₂O₃-modified copper reveals that its resistive properties could be a universal recipe for dendrite control. It is worth noting that even though resistive films outperform conductive films at lower current densities,

the overpotential penalties associated with resistive films could limit their application at higher current densities.

4) What is the effect of the single defect size of the resistive nanofilm on the performance of Li/Cu or anode-free Cu||NMC 532 cells. Does the total surface area of all defects affect the performance?

Response: We anticipate that an increase in the size of defects will increase the number of Li nuclei that form on the bare copper. And because nucleation on bare copper results in higher surface area lithium particles, we expect that electrolyte consumption will increase with bigger defect sizes, as we will be approaching a current collector with the properties of bare copper. Given the scope of this work, we focused on using small defects to illustrate the concept of ion diffusion and radial growth of lithium, so the exact relationship between defect size and battery performance is the subject of ongoing investigations in our lab.

5) The citation format of references should be consistent. For example, reference 11 has DOI number and should be modified.

Response: We have modified our references for consistency. We are grateful to the reviewer for pointing this out.

References

1. Weber, R. *et al.* Long cycle life and dendrite-free lithium morphology in anode-free lithium pouch cells enabled by a dual-salt liquid electrolyte. *Nat. Energy* **4**, 683–689 (2019).
2. Zheng, J. *et al.* Electrolyte additive enabled fast charging and stable cycling lithium metal batteries. *Nat. Energy* **2**, 17012 (2017).
3. Ren, X. *et al.* Enabling High-Voltage Lithium-Metal Batteries under Practical Conditions. *Joule* **3**, 1662–1676 (2019).
4. Yu, Z. *et al.* Molecular design for electrolyte solvents enabling energy-dense and long-cycling lithium metal batteries. *Nat. Energy* **5**, 526–533 (2020).
5. Sun, Y. *et al.* A Novel Organic “Polyurea” Thin Film for Ultralong-Life Lithium-Metal Anodes via Molecular-Layer Deposition. *Adv. Mater.* **31**, 1806541 (2019).
6. Zhao, Y. *et al.* Robust Metallic Lithium Anode Protection by the Molecular-Layer-Deposition Technique. *Small Methods* **2**, 1700417 (2018).
7. Zhao, Y. *et al.* Natural SEI-Inspired Dual-Protective Layers via Atomic/Molecular Layer Deposition for Long-Life Metallic Lithium Anode. *Matter* (2019).
8. Kozen, A. C. *et al.* Next-Generation Lithium Metal Anode Engineering via Atomic Layer Deposition. *ACS Nano* **9**, 5884–5892 (2015).
9. Kazyak, E., Wood, K. N. & Dasgupta, N. P. Improved Cycle Life and Stability of Lithium Metal Anodes through Ultrathin Atomic Layer Deposition Surface Treatments. *Chem. Mater.* **27**, 6457–

- 6462 (2015).
10. Wang, J. *et al.* Improving cyclability of Li metal batteries at elevated temperatures and its origin revealed by cryo-electron microscopy. *Nat. Energy* 2019 48 **4**, 664–670 (2019).
 11. Yu, Z. *et al.* Molecular design for electrolyte solvents enabling energy-dense and long-cycling lithium metal batteries. *Nat. Energy* 2020 57 **5**, 526–533 (2020).
 12. Chen, X. *et al.* A Diffusion- - Reaction Competition Mechanism to Tailor Lithium Deposition for Lithium- Metal Batteries. *Angew. Chemie* **132**, 7817–7821 (2020).
 13. Huang, W. *et al.* Nanostructural and Electrochemical Evolution of the Solid-Electrolyte Interphase on CuO Nanowires Revealed by Cryogenic-Electron Microscopy and Impedance Spectroscopy. *ACS Nano* **13**, 737–744 (2019).
 14. Shi, F. *et al.* Strong texturing of lithium metal in batteries. *Proc. Natl. Acad. Sci. U. S. A.* **114**, 12138–12143 (2017).
 15. Peled, E. & Menkin, S. Review—SEI: Past, Present and Future. *J. Electrochem. Soc.* **164**, A1703–A1719 (2017).
 16. Xiao, J. *et al.* Understanding and applying coulombic efficiency in lithium metal batteries. *Nat. Energy* **5**, 561–568 (2020).
 17. Lin, D. *et al.* Fast galvanic lithium corrosion involving a Kirkendall-type mechanism. *Nat. Chem.* doi:10.1038/s41557-018-0203-8.
 18. Kolesnikov, A. *et al.* Galvanic Corrosion of Lithium-Powder-Based Electrodes. *Adv. Energy Mater.* **10**, 2000017 (2020).
 19. Cevher, O. & Akbulut, H. Electrochemical performance of SnO₂ and SnO₂/MWCNT/graphene composite anodes for Li-Ion batteries. *Acta Phys. Pol. A* **131**, 204–206 (2017).
 20. Ferraresi, G. *et al.* SnO₂ Model Electrode Cycled in Li-Ion Battery Reveals the Formation of Li₂SnO₃ and Li₈SnO₆ Phases through Conversion Reactions. *ACS Appl. Mater. Interfaces* **10**, 8712–8720 (2018).
 21. Shen, S., Zhong, W., Huang, X., Lin, Y. & Wang, T. Ordered ZnO/Ni hollow microsphere arrays as anode materials for lithium ion batteries. *Materials (Basel)*. **12**, (2019).
 22. Liu, Y. *et al.* An Artificial Solid Electrolyte Interphase with High Li-Ion Conductivity, Mechanical Strength, and Flexibility for Stable Lithium Metal Anodes. *Adv. Mater.* **29**, 1–8 (2017).
 23. Shi, F. *et al.* Strong texturing of lithium metal in batteries. *Proc. Natl. Acad. Sci. U. S. A.* **114**, 12138–12143 (2017).
 24. Jaumann, T. *et al.* Role of 1,3-Dioxolane and LiNO₃ Addition on the Long Term Stability of Nanostructured Silicon/Carbon Anodes for Rechargeable Lithium Batteries. *J. Electrochem. Soc.* **163**, A557–A564 (2016).
 25. Single, F., Horstmann, B. & Latz, A. Theory of Impedance Spectroscopy for Lithium Batteries. *J. Phys. Chem. C* (2019).
 26. Thirumalraj, B. *et al.* Nucleation and Growth Mechanism of Lithium Metal Electroplating. (2019).
 27. Pei, A., Zheng, G., Shi, F., Li, Y. & Cui, Y. Nanoscale Nucleation and Growth of Electrodeposited Lithium Metal. *Nano Lett* **17**, 1 (2017).

28. Boyle, D. T. *et al.* Corrosion of lithium metal anodes during calendar ageing and its microscopic origins. *Nat. Energy* **6**, 487–494 (2021).
29. Li, Y. *et al.* Atomic structure of sensitive battery materials and interfaces revealed by cryo-electron microscopy. *Science (80-.)*. **358**, 506–510 (2017).
30. Yu, Z. *et al.* Molecular design for electrolyte solvents enabling energy-dense and long-cycling lithium metal batteries. *Nat. Energy* **5**, 526–533 (2020).
31. Fang, C. *et al.* Quantifying inactive lithium in lithium metal batteries. *Nature* **572**, 511–515 (2019).
32. Zheng, J. *et al.* Physical Orphaning versus Chemical Instability: Is Dendritic Electrodeposition of Li Fatal? *ACS Energy Lett.* **4**, 1349–1355 (2019).
33. Oyakhire, S. T. *et al.* Revealing and Elucidating ALD- Derived Control of Lithium Plating Microstructure. *Adv. Energy Mater.* 2002736 (2020).
34. Hu, B., Yao, M., Xiao, R., Chen, J. & Yao, X. Optical properties of amorphous Al₂O₃ thin films prepared by a sol-gel process. *Ceram. Int.* **40**, 14133–14139 (2014).
35. Wang, J. *et al.* Fundamental study on the wetting property of liquid lithium. *Energy Storage Mater.* **14**, 345–350 (2018).
36. Kwoka, M., Ottaviano, L., Passacantando, M., Santucci, S. & Szuber, J. XPS depth profiling studies of L-CVD SnO₂ thin films. *Appl. Surf. Sci.* **252**, 7730–7733 (2006).
37. Lu, Y. F., Ni, H. Q., Mai, Z. H. & Ren, Z. M. The effects of thermal annealing on ZnO thin films grown by pulsed laser deposition. *J. Appl. Phys.* **88**, 498–502 (2000).
38. Ou, G. *et al.* Tuning defects in oxides at room temperature by lithium reduction. *Nat. Commun.* **9**, 1–9 (2018).
39. Zhang, Y. *et al.* Investigation of the defect density in ultra-thin Al₂O₃ films grown using atomic layer deposition. *Surf. Coatings Technol.* **205**, 3334–3339 (2011).
40. Klumbies, H. *et al.* Thickness dependent barrier performance of permeation barriers made from atomic layer deposited alumina for organic devices. *Org. Electron.* **17**, 138–143 (2015).
41. Litvinov, J. *et al.* Development of pinhole-free amorphous aluminum oxide protective layers for biomedical device applications. *Surf. Coatings Technol.* **224**, 101–108 (2013).

Reviewer comments, second round review

Reviewer #1 (Remarks to the Author):

The manuscript is timely, original and of great importance. The work supports the conclusions and claims and provides new insights on lithium plating. The methodology is consistent, and the findings are strongly supported.

I recommend accepting the manuscript as is. I would like to thank the authors for conducting additional experiments and addressing all my comments and questions. The revised manuscript is excellent, and I believe the scientific community will benefit from it greatly.